# AI-based chest X-ray prioritization in the lung cancer diagnostic pathway: the LungIMPACT randomized controlled trial

Prioritizing artificial intelligence (AI)-detected imaging findings may reduce the time to diagnosis of lung cancer. This prospective, multicentre, randomized controlled trial tested whether immediate AI prioritization of primary care-requested chest X-rays (CXR) influenced time to computed tomography (CT) and lung cancer diagnosis, the primary outcomes. Secondary outcomes included the number of urgent suspected lung cancer referrals, incidence and stage of lung cancer, times to urgent referral and treatment, concordance between AI and radiology reports, and algorithm accuracy. AI was available in both study arms, with AI prioritization randomized by day. Of 97,731 participant CXRs, 4,405 were excluded due to data compliance issues or failure of randomization, resulting in 93,326 CXRs analyzed (45,987 and 47,339 in the prioritization 'on' or 'off' arms, respectively). A total of 13,347 CTs were identified, with 2,766 performed within 14 days of CXR. Median (interquartile range) times to CT were 53 days (17–145) and 53 days (19–141), with and without AI prioritization, corresponding to a ratio of geometric means of 0.97 (95% confidence interval (CI) = 0.93–1.02; $P$ = 0.31). When restricted to CTs performed within 14 days of CXR, the median time to CT was 8 days (5–11) in both groups. Lung cancer was diagnosed in 558 people (0.6% of CXRs). Median times to diagnosis were 44 days (26–90) and 46 days (24–105) respectively, with a ratio of geometric means of 0.98 (95% CI = 0.83–1.16; $P$ = 0.84). No significant differences were observed in time to lung cancer referral (14 versus 15 days; $P$ = 0.13), time to treatment (76 versus 72.5 days; $P$ = 0.99) or stage at diagnosis ($P$ = 0.34). Discordance between AI and radiology reports occurred in 28,261 CXRs (30.3%) and expert radiology review identified actionable findings in 6,750 cases (23.9%). AI prioritization of CXR requested by UK primary care has no significant impact on the lung cancer pathway. Therefore, CXR AI deployments should not include worklist prioritization in this context. Future research should differentiate between primary pathway changes and the direct impact of AI. ISRCTN registration: 78987039.

✉e-mail: david.baldwin12@nhs.net

Lung cancer accounts for the highest proportion of cancer-related deaths in the United Kingdom (UK) and worldwide, primarily because it is common, is diagnosed at a late stage and/or late in the symptomatic period when patients have become too unwell for treatment. Survival rates in the UK are lower than those from high-income countries with comparable cancer registration data and it is thought that much of this is due to later diagnosis[1–3]. Diagnostic pathway delay can worsen outcomes in both early-stage and late-stage disease and increase patient and carer anxiety[4–6]. Patient feedback gathered from focus groups conducted during the study also supports the need for timely reporting. Patients who undergo investigations are often anxious about the results, with research suggesting that the time between testing and receiving the results is particularly worrying[7–10].

National cancer care pathways provide a method to accelerate and standardize the end-to-end patient journey[11,12]. In England, the National Optimal Lung Cancer Pathway (NOLCP) recommends rapid progression from chest X-ray (CXR) to computed tomography (CT), followed by assessment in a specialist clinic. The NOLCP mandates that patients with a suspicious CXR undergo CT within 72 h, preferably on the same day. This aims to reduce delays experienced by patients, including those referred through alternative routes[12]. A single attendance for initial diagnostic investigations may reduce delays in subsequent appointments and eliminate the chance of communication not being received or understood. Recently published research funded by Cancer Research UK, conducted by members of the current study team, found that immediate radiographer CXR reporting, with direct triage to same-day CT, significantly reduced time to diagnosis of lung cancer by almost half from a median of 63 days from CXR, to 32 days ($P = 0.03$) compared to routine CXR reporting once the patient had left the department[13]. This demonstrated that immediate CXR reporting and prioritization are feasible and may accelerate early symptomatic diagnosis of lung cancer, which remains a major challenge despite clear referral criteria from the National Institute for Health and Care Excellence (NICE)[14,15]. Current UK guidance also recommends increasing the use of CXR, as there is some evidence that this improves survival and the proportion of early-stage lung cancer diagnoses[15–18].

Artificial intelligence (AI) algorithms for assisted reporting of CXRs have been proposed to improve both the speed and accuracy of diagnosis and to help the overstretched radiology workforce. However, a recent review of the clinical utility of this technology by NICE concluded that there was insufficient evidence to make any recommendations other than to ensure products were carefully evaluated[19]. NICE cited the LungIMPACT study as the only investigation considered potentially capable of addressing important clinical questions[19]. The primary aim of the trial was to measure the impact of immediate AI-driven prioritization of abnormal CXRs for reporting on the time to CT and diagnosis of lung cancer. Secondary outcomes included impact on other steps in the pathway and detailed discordance reviews of AI in both arms.

## Results

### Descriptive statistics and consort diagram
Between 17 July 2023 and 31 December 2024, a total of 97,731 CXRs were performed across five diverse National Health Service (NHS) Trusts, with a geographical spread and a mix of general and tertiary, as well as high-volume and low-volume, centers. Patients were followed up until 5 June 2025. Figure 1 shows the CONSORT diagram of the study population. After data cleaning, 93,326 CXRs from 86,945 patients were included in the study analysis. Of these, 45,987 (49.3%) CXRs were performed in sessions with AI prioritization for immediate review.

Table 1 presents the characteristics and demographics of the study population stratified by AI prioritization day. The mean age of the study population was 59 years and 46% were male. University Hospitals of Leicester performed 54% of the study CXRs, Nottingham

University Hospitals 15%, East Sussex and North Essex Foundation Trust 11%, University College London Hospital 11% and University Hospitals of Birmingham 9%.

### Primary outcomes
Among all patients, 13,347 had a valid CT scan and were included in the analysis, with a time window from CXR acquisition to CT scan analysis—6,674 with AI prioritization and 6,673 with no AI prioritization. A total of 558 patients were diagnosed with lung cancer and were included in the analysis of time to cancer diagnosis, 269 with AI prioritization and 289 with no AI prioritization.

The median time to CT scan for all patients was 53 days (interquartile range (IQR) = 17–145 days) for AI prioritization days and 53 days (IQR = 19–141 days) for those with no AI prioritization. Similarly, the median time to lung cancer diagnosis was 44 days (IQR = 26–90 days) for AI prioritization days and 46 days (IQR = 24–105 days) for those with no AI prioritization (Table 2). There was no significant difference in the time from CXR acquisition to CT scan according to AI prioritization, with a ratio of geometric means of 0.97 (95% confidence interval (CI) = 0.93–1.02, $P = 0.31$). There was no difference in time to CT when restricted to CT scans performed within 14 days and CXRs with a CT referral ($P = 0.96$ and $P = 0.86$, respectively; Table 2). There was no significant difference in the time from CXR acquisition to lung cancer diagnosis between AI prioritization days, with a ratio of geometric means of 0.98 (95% CI = 0.83–1.16, $P = 0.84$; Table 2). No statistically significant differences were found for either primary outcome at any site (Extended Data Table 1) or for any quarter (Extended Data Table 2). Primary outcomes by sex are presented in Extended Data Table 4.

### Secondary outcomes
The median time to two-week-wait (2WW) referral was 14 days (IQR = 5–53 days) for AI prioritization days and 15 days (IQR = 6–50 days) for those with no AI prioritization. The median time to initiation of cancer treatment was 76 days (IQR = 38–114 days) for AI prioritization days and 73 days (IQR = 43–121 days) for those with no AI prioritization (Table 2). Extended Data Table 3 presents the stage and incidence of lung cancer in each arm. There were no differences in secondary outcomes by AI group.

Table 3 presents the absolute numbers of true-positive (TP), false-positive (FP), true-negative (TN) and false-negative (FN) CXRs and the number of subsequent cancers diagnosed within each group after discordance reviews. Discordance reviews were completed for 26,505 of 28,261 discordant CXR reports (30.3%) of all CXRs. In 1,756 cases, discordance review was not performed because the category was not considered potentially serious and the prespecified limit of 1,000 reviews per category had been reached. The sensitivity and specificity relate to the radiology concordance review of the CXR, which was regarded as the reference standard. Table 3 shows a total of 254,349 discordant findings reviews; 5.5% were TP, 11.6% FP, 2.3% FN and 81.2% TN. There were 387 subsequent cancers diagnosed. More than one abnormality was often present on the CXR and patients with lung cancer may have had more than one CXR, so each lung cancer may be associated with multiple abnormality classifications. AI was positive for 354 of 387 cancers in the opacity category and 215 of 387 in the nodule category. However, 107 of 11,284 opacity classifications and 150 of 5,783 nodule classifications were judged to be FPs. AI was negative for nodule and cavity category in 20 of 343 and 9 of 1,395 CXRs, respectively, that subsequently resulted in a diagnosis of lung cancer. The agreement between AI and the radiology report for each classification is presented in Extended Data Table 5. Table 4 shows the recommendations made following expert discordance review with the number of lung cancers diagnosed in the period following the CXR. Note that this table shows recommendations after discordance for any findings category.

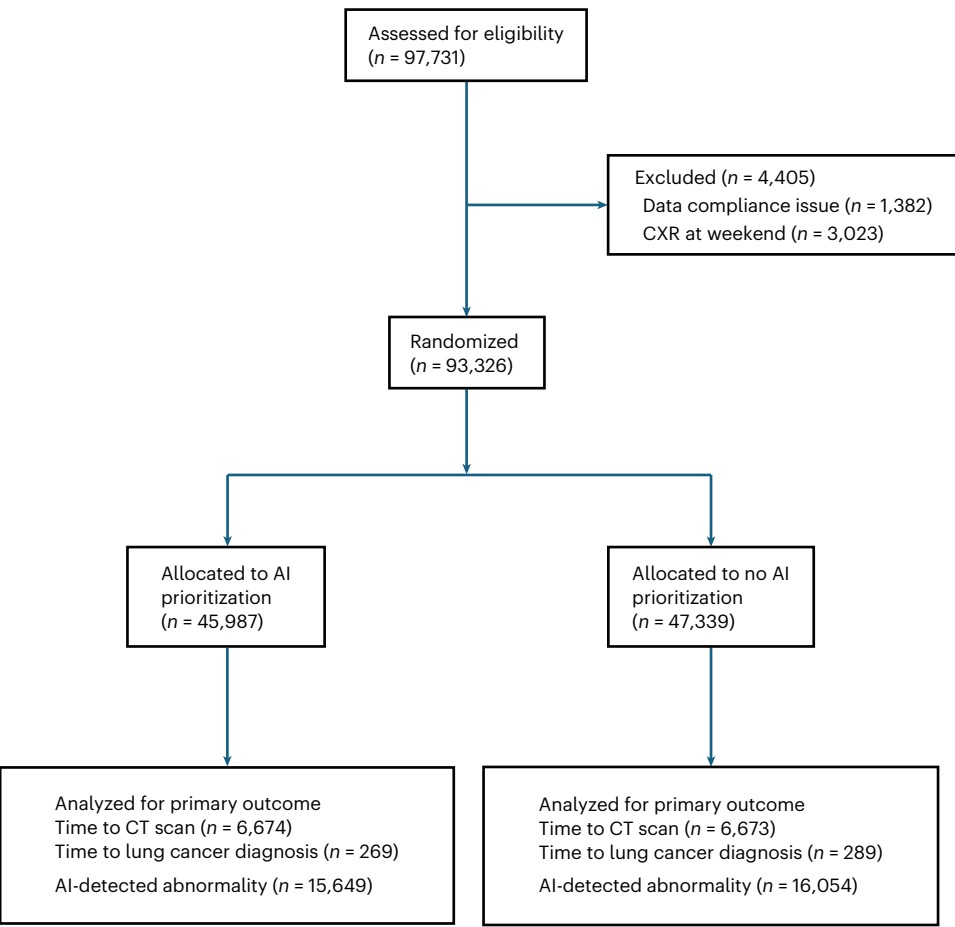

**Fig. 1 | Study population for CXRs CONSORT diagram.** The total number of CXRs assessed for eligibility, the numbers analyzed for the primary outcomes, and the number of AI discordances in each arm of the study.

## Post hoc analyses

**Patients with a confirmed diagnosis of cancer.** For patients diagnosed with cancer, the time to CT and time to cancer diagnosis were longest when both the radiologist and AI CXR reports were normal; median of 72 days to CT and median of 177 days to cancer diagnosis, compared to a median of 8 days to CT and 38 days to cancer diagnosis with both the radiologist and AI CXR reports were abnormal for any reason (Table 5). These differences were statistically significant, with the time to CT scan seven times higher ($P < 0.001$) and the time to cancer diagnosis three times higher ($P < 0.001$) for the radiologist and AI CXR reports both normal compared to both the radiologist and AI CXR reports abnormal (Table 5). Of the total of 558 cancers, 53 were in the radiology-normal/AI-abnormal group, with a median time to diagnosis of 106 days—potentially within a window when CT would have been able to diagnose early-stage cancer and 2.27 times longer geometric mean than when both radiology and AI were abnormal (median time to diagnosis of 38 days). In contrast, the ratio of geometric means was only 1.24:1 for the 22 cancers in the radiologist-abnormal/AI-normal group (median time to diagnosis of 50 days). Duration of patients with a lung cancer diagnosis for AI prioritization on and off is visualized in Extended Data Fig. 1.

**Timings relevant to the NOLCP.** A total of only 172 CT scans were completed on the same day as the CXR, with 88 in the prioritization 'on' and 84 in the prioritization 'off' arms. This is the recommended best practice in the NOLCP. There were 477 CT scans within the maximum time of 72 h from CXR as set out in the NOLCP, with 241 in the prioritization 'on' and 236 in the prioritization 'off' arms.

## Discussion

This large, multisite randomized study has shown that AI prioritization of CXRs did not improve the speed of the lung cancer diagnostic pathway. This means that this element of AI functionality, which introduces additional complexity and cost to AI installation and may add time to clinical workflows, is not required to accelerate the lung cancer diagnostic pathway. A detailed health economics evaluation will be published separately, but costs are both considerable and avoidable, given the results reported in this study. Several other important findings from the secondary outcomes and the exploratory analyses are discussed below.

CXR, often the first test for patients with lung cancer referred for diagnosis, is the most frequently performed imaging investigation, with more than 7 million performed annually in England, including ~2.2 million CXRs referred from primary care alone[20]. Following either an abnormal CXR or presentation in a high-risk patient, the NOLCP mandates rapid progression to CT. Current (April 2025) median reporting times for CXRs referred from primary care (2 days) and waiting time for CT chest (15 days) do not meet the NOLCP standards (72 h from abnormal CXR to CT)[20]. The lower number of CT scanners per capita[21], as well as chronic shortage of consultant radiologists and radiographers and an ever-increasing workload, may explain the lack of compliance.

One potential strategy to optimize progression from CXR to CT and bring forward the diagnosis is to use an AI tool to prioritize CXRs with suspicious findings for urgent reporting and improve the accuracy of interpretation, thereby selecting patients most likely to benefit from a same-day CT scan. However, this approach needs careful evaluation, as AI might have the opposite effect of not increasing accuracy or

**Table 1 | Study population (CXR) by AI prioritization**

| Study population | AI prioritization day | | |
|---|---|---|---|
| | **No** | **Yes** | **Total** |
| | **(n = 47,339)** | **(n = 45,987)** | **(n = 93,326)** |
| Age (years) | 59.1 (17.4) | 59.1 (17.5) | 59.1 (17.4) |
| Sex | | | |
| Male | 21,987 (46.4%) | 21,278 (46.3%) | 43,265 (46.4%) |
| Female | 25,352 (53.6%) | 24,709 (53.7%) | 50,061 (53.6%) |
| Hospital trust | | | |
| University College London Hospital | 5,216 (11.0%) | 5,360 (11.7%) | 10,576 (11.3%) |
| University Hospitals of Leicester | 25,772 (54.4%) | 24,287 (52.8%) | 50,059 (53.6%) |
| East Sussex and North Essex Foundation Trust | 5,508 (11.6%) | 5,148 (11.2%) | 10,656 (11.4%) |
| University Hospitals of Birmingham | 3,878 (8.2%) | 4,064 (8.8%) | 7,942 (8.5%) |
| Nottingham University Hospitals | 6,965 (14.7%) | 7,128 (15.5%) | 14,093 (15.1%) |
| Quarter | | | |
| Q3 2023 | 597 (1.3%) | 632 (1.4%) | 1,229 (1.3%) |
| Q4 2023 | 6,127 (12.9%) | 5,850 (12.7%) | 11,977 (12.8%) |
| Q1 2024 | 9,326 (19.7%) | 9,839 (21.4%) | 19,165 (20.5%) |
| Q2 2024 | 12,986 (27.4%) | 12,277 (26.7%) | 25,263 (27.1%) |
| Q3 2024 | 8,876 (18.7%) | 8,535 (18.6%) | 17,411 (18.7%) |
| Q4 2024 | 9,427 (19.9%) | 8,854 (19.3%) | 18,281 (19.6%) |

Values are mean (s.d.) or n (%).

speed and creating extra workload, for example, through FP findings or altered clinician behavior. Vigilance fatigue is of particular concern, that is, when the review of all CXRs flagged as abnormal either desensitizes the reader to potential subsequent FN CXRs labeled as normal by the AI or causes the reader to lose trust in the AI's accuracy (the 'cry wolf' phenomenon)[22]. This is one reason why NICE has not recommended any AI products for CXR interpretation in England[23].

The LungIMPACT study has examined both the above elements of CXR use through its primary outcomes and several other aspects with the secondary outcomes, including analysis of discordances and the likelihood of a cancer diagnosis with the addition of AI. It has been shown conclusively that AI prioritization had no impact on time to CT or time to lung cancer diagnosis. In the secondary outcome analysis, further steps of the pathway were examined; the study has quantified the number of FPs and FNs and shown the number of lung cancers expected to be diagnosed (Tables 3 and 4). Exploratory analyses also showed differences in time to CT and time to lung cancer diagnosis across the different combinations of discordance (Table 5).

A significant reduction in the median time from CXR acquisition to report was observed, from 47 h to 34.1 h; however, no significant differences were detected in any of the timings measured as primary or secondary outcomes (Table 2). The lack of improvement in AI prioritization is likely because, although the time to CXR report was shortened, it was not sufficiently reduced in the prioritization arm to influence the clinical pathway in the same way as immediate radiographer triage did in our previous study[13]. This may, in turn, be explained by the limited capacity within Trusts to both report immediately and organize downstream tests and appointments, even though this is a recommendation in the NOLCP. However, this finding could not have been predicted with certainty because our previous study showed a marked effect on time

to diagnosis under similar service constraints, albeit on a smaller scale. It could be argued that pathway change should have been mandated in the prioritization arm to ensure all prioritized CXRs were reviewed by a radiologist before the patient left the department. However, this represents a pathway change that may have an impact on its own and has little to do with AI-based prioritization (see review of other studies below). Furthermore, as shown in Extended Data Fig. 2, immediate review of the CXR was permitted, according to local procedure, in both arms of the study. This ensured that AI prioritization was tested rather than the effect of primary pathway change.

A large number of discordance reviews were completed (26,505), representing 28.4% of the randomized CXRs and 94% of all discordances. Although results are presented for eight classifications, the 'other' category includes an additional nine findings (none of which were relevant to lung cancer). Expert radiology review was considered the gold standard. Compared with radiologists, AI FP findings comprised 11.6% of the total across all categories. Some of these will be from the same CXR (proportion of FP plus FN of all reviewed CXRs was 33,910/28,261; ratio = 1.2:1). However, feedback from reporters indicated that most of the FPs are rapidly dismissed, so the potential for extra workload will be dependent on the user becoming familiar with the strengths and weaknesses of the AI. An important finding from our study relates to the number of cancers subsequently diagnosed in the cohort with a positive AI finding that was dismissed by the radiologist or reporting radiographer and the apparent association with time to diagnosis. We are aware of only a few cases in which there was a definite error on behalf of the reporter after review and this merits a detailed evaluation of the identified cases to establish whether they can be appropriately managed, perhaps using a combination of baseline risk[24] and CXR AI findings. It is also important to establish the reason for FNs. These were relatively few in number, with over half in the 'other' and cardiomegaly categories (3,366/5,777). However, for the nodule category, with only 343 AI FNs, 20 cancers were diagnosed, representing 5% of the cancers in that category (Table 3).

A major strength of the trial is its randomized controlled design, which does not require obtaining individual consent from participants. This meant minimal influence on the normal clinical pathway and more equitable inclusion of groups underrepresented in research. Information provided in the radiology departments indicated that participants could opt out of the study; however, only one participant did so. In addition, the trial was multicentre, encompassing NHS hospitals of different sizes and catchment demographics. The automated data acquisition followed by manual verification is another strength of the study and the independent analysis of the results with no influence from either the AI developers or the investigators. Another major strength is the number of discordance reviews completed by expert radiologists and reporting radiographers. This has provided perhaps the largest reliable dataset on this aspect. Review/recall methodology is a form of radiology peer review, advocated as an effective method of quality assurance[25–29], but it is seldom, if ever, done on this scale.

A strength of the trial is also the use of two clinically meaningful, real-world, primary outcomes: time to lung cancer diagnosis and time to CT. Both are highly relevant to the goal of achieving earlier diagnosis in lung cancer, but the former is more dependent on the many steps involved in the pathway. In addition, secondary outcomes provided insight into other potential impacts, including the number of lung cancer diagnoses, the stage at diagnosis and the number of urgent referrals, which in our previous work showed trends towards being more favorable in the immediate reporting arm, but with too few cancers to allow a statistically significant result (n = 49)[13].

A limitation of the study is that the primary outcomes evaluated only the impact of AI prioritization, not the presence or absence of AI in the clinical workflow. This was necessary because of the study's nonconsenting nature. Consenting would have been impractical and could itself have influenced the primary outcome through

**Table 2 | Time outcomes by AI prioritization, median and IQR**

| Outcomes | AI prioritization | | | | | | | 95% CI | P value |
|---|---|---|---|---|---|---|---|---|---|
| | No | | | Yes | | | | | |
| | (n=44,112)ᵃ | | | (n=42,833)ᵃ | | | | | |
| | n | Median | IQR | n | Median | IQR | Ratio of geometric means | | |
| Primary outcomes | | | | | | | | | |
| Time to CT scan (days) | | | | | | | | | |
| All CT scans | 6,673 | 53 | (19–141) | 6,674 | 53 | (17–145) | 0.97 | (0.93–1.02) | 0.31 |
| CT scans within 14 days of CXR | 1,314 | 8 | (5–11) | 1,452 | 8 | (5–11) | 1.00 | (0.91–1.10) | 0.96 |
| CXRs coded to generate a CT referral | 495 | 7 | (4–9) | 505 | 6 | (4– 9) | 1.02 | (0.81–1.28) | 0.86 |
| Time to lung cancer diagnosis (days) | 289 | 46 | (24–105) | 269 | 44 | (26– 90) | 0.98 | (0.83–1.16) | 0.84 |
| Secondary outcomes | | | | | | | | | |
| Time to 2WW referral (days) | 1,284 | 15 | (6–50) | 1,215 | 14 | (5–53) | 0.91 | (0.80–1.03) | 0.13 |
| Time from CXR to CXR report (h) | 44,078 | 47.0 | (15.8–99) | 42,814 | 34.1 | (6.6–93.1) | 0.85 | (0.83–0.87) | <0.001 |
| Time to cancer treatment starting (days) | 200 | 72.5 | (43–120.5) | 200 | 76 | (37–114) | 1.00 | (0.84–1.19) | 0.99 |

Statistical analyses were performed using a two-sided *t*-test on log-transformed outcomes and presented as the ratio of geometric means and 95% CIs. ᵃOnly includes the first CXR for patients with multiple CXRs in the study period.

**Table 3 | Review outcomes and cancers diagnosed by abnormality for CXRs with review (numbers in parentheses indicate the number of cancers diagnosed within each group)**

| Abnormality category | True positive | False positive | False negative | True negative | Sensitivity | Specificity |
|---|---|---|---|---|---|---|
| Nodule | 1,306 (65) | 5,783 (150) | 343 (20) | 20,829 (152) | 0.79 | 0.78 |
| Cavity | 40 (6) | 257 (14) | 35 (5) | 27,929 (362) | 0.53 | 0.99 |
| Mediastinal widening | 73 (2) | 667 (3) | 182 (17) | 27,339 (365) | 0.29 | 0.98 |
| Hilar enlargement | 186 (13) | 2,997 (24) | 169 (33) | 24,909 (317) | 0.52 | 0.89 |
| Blunted CP/pleural effusion | 1,935 (65) | 2,827 (52) | 284 (4) | 23,215 (266) | 0.87 | 0.89 |
| Opacity | 7,931 (247) | 11,284 (107) | 1,395 (9) | 7,651 (24) | 0.85 | 0.40 |
| Cardiomegaly | 1,425 (5) | 2,141 (18) | 799 (4) | 23,896 (360) | 0.64 | 0.92 |
| Pneumothorax | 11 (0) | 447 (8) | 3 (0) | 27,800 (379) | 0.79 | 0.98 |
| Other | 1,068 (19) | 1,730 (48) | 2,567 (29) | 22,896 (291) | 0.29 | 0.93 |
| Total | 13,975 | 28,133 | 5,777 | 206,464 | – | – |

greater-than-normal scrutiny of imaging within the context of the busy NHS. Furthermore, the use of AI, rather than no use at all, could improve triage quality and potentially the speed of diagnosis. However, it would likely have less influence on CT requests unless the AI were highly accurate. By choosing time to CT as an additional primary outcome, we can be more confident that a comparison of AI use versus no AI would not show a meaningful difference.

A limitation is the use of a single AI product, which will inevitably have different performance characteristics from others. This issue is likely to persist as algorithms are updated. However, the fact that only prioritization was tested here means differences in AI accuracy had less effect on the coprimary outcomes than on the discordances.

A further limitation of the study was the difficulty in identifying the best way to measure time to CT. Including all CTs was necessary to ensure that any potential adverse effect of AI prioritization, such as 'reassurance' from a normal AI finding, were captured. This was

because we found that many patients having CXRs go on to have CTs, which may have no relationship to the CXR—for example, due to subsequent respiratory assessment or emergency department attendance, even many days after the first referral. While this finding is of interest in itself, it may have diluted this primary outcome by adding many CTs that were not relevant to the hypothesis. This explains why the time to CT, for all downstream scans, was so long (indeed longer than the time to lung cancer diagnosis). We therefore had to complete restrictive sensitivity analyses using the code, where used, signifying a CT was triggered and also by limiting the time after CXR to less than 14 days, having determined that these were likely related to the initial CXR. This was the case even when the CXR was normal, but the risk remained high enough to warrant referral for subsequent CT. Despite this limitation, the results clearly show no difference in time from CXR to CT according to prioritization day across all analyses, each of which included a more than adequate number of observations (Table 3).

**Table 4 | Recommendation after expert discordance review**

| Outcome recommendation | n (%) | Number of cancers diagnosed |
|---|---|---|
| No actionable finding | 21,511 (76.1%) | 57 |
| Suspected lung cancer—refer for CT and to lung cancer MDT | 672 (2.4%) | 232 |
| Pulmonary nodule requiring CT | 559 (2.0%) | 52 |
| Suspected other cancer—refer to other cancer MDT | 52 (0.2%) | 5 |
| Incidental thoracic finding—requires respiratory review | 488 (1.7%) | 11 |
| Incidental nonthoracic finding—requires other secondary care action | 146 (0.5%) | 0 |
| Incidental findings for primary care action | 3,077 (10.9%) | 21 |

Number of CXRs reviewed n = 26,505. Note that no review was undertaken for 1,756 discordances. For common and less serious categories, a maximum of 1,000 reviews was required. MDT, multidisciplinary team.

**Table 5 | For those diagnosed with cancer, time to CT scan and time to cancer diagnosis by radiologist and AI CXR report normal/abnormal, median days and ratio of geometric means**

| CXR report agreement | n | Median days (IQR) | Ratio of geometric means | 95% CI | P value |
|---|---|---|---|---|---|
| Time to CT | | | | | |
| Radiology and AI normal | 30 | 72 (32–154) | 7.35 | (3.63–14.9) | <0.001 |
| Radiology normal AI abnormal | 51 | 46 (9–121) | 4.70 | (2.70–8.17) | <0.001 |
| Radiology abnormal and AI normal | 20 | 15 (7–22.5) | 1.59 | (0.68–3.73) | 0.29 |
| Radiology and AI abnormal | 446 | 8 (4–21) | 1.0 | – | – |
| Time to cancer diagnosis | | | | | |
| Radiology and AI normal | 33 | 177 (91–227) | 3.43 | (2.47–4.75) | <0.001 |
| Radiology normal AI abnormal | 53 | 106 (52–187) | 2.27 | (1.75–2.96) | <0.001 |
| Radiology abnormal and AI normal | 22 | 50 (30–75) | 1.24 | (0.83–1.84) | 0.29 |
| Radiology and AI abnormal | 450 | 38 (22–72) | 1 | – | – |

Statistical analyses were performed using linear regression on log-transformed outcomes and presented as the ratio of geometric means and 95% CIs. All hypothesis testing was conducted using two-sided tests.

Another limitation relates to the discordance reviews, which have been reported according to differences between the AI flag classification and the final report. Because reporters had the AI available at the time, we cannot determine how many findings would be independently found by the AI or whether the human would have found something without AI assistance. However, the purpose of the discordance review was merely to confirm that the reports were indeed discordant, and this is what is presented. There were a small number of discordances in which the AI was correct and the reporter incorrect; these have not yet been fully quantified or analyzed. These cases, together with a detailed analysis of downstream outcomes, are the subject of a further project.

In England, there has been a phased implementation of lung cancer screening for people aged 55 to 74 at high risk of developing lung cancer. From Table 1 and Extended Data Table 1, the cancer rate per patient can be calculated, demonstrating that University College London Hospital, which has an established screening program, had a rate of 0.38%, whereas University Hospitals of Birmingham, which commenced screening in 2022, had a rate of 0.53%. However, the rate for University Hospitals of Leicester (without a program) and Nottingham University Hospitals (program started in 2023) was 0.7%. Our study is unable to determine whether the screening program is reducing the early-stage detection by CXR.

The health economics analysis is underway and considers the comparison made in the study, as well as focusing on the differences between using AI at all and not using AI, in a planned secondary analysis.

Research on the impact of AI prioritization of the CXR is very limited. A prospective study is currently underway in Glasgow, Scotland, assessing the impact of AI prioritization on time to CT among patients with lung cancer[30]; this is the only other randomized prospective study we are aware of.

The research on the impact of CXR AI on clinical outcomes and accuracy is also limited. Some information comes from company product specifications, which show favorable area under the curve (AUC) values across a variety of classifications[31,32]. In a retrospective study of 6,006 CXRs from a respiratory department, AI assistance improved radiologists' detection performance for nodules/masses, consolidation and pneumothorax, increasing the AUC from 0.861 to 0.886 (P = 0.003; ref. 33). Another retrospective study using 2,568 CXRs acquired in a range of settings showed an AUC for radiologists of 0.71, which improved to 0.81 with AI assistance, but also noted a remarkable AUC of 0.96 with the AI model alone[34]. A further retrospective study of 563 emergency unit CXRs showed improvement in the accuracy of imaging diagnosis among nonradiology residents[35]. A small, single site (n = 68 lung cancers), nonrandomized before-and-after 12-month service evaluation conducted in Scotland with another CXR AI algorithm could not attribute the small (mean 51 versus 58 days) reduction in time to treatment to the use of AI as multiple pathway changes were made (additional radiologist, administrative support, pathway navigator) and resulted in an increase in cost of £3.59 per CXR[36]. Another single-trust before-and-after comparison of AI found a shorter time from CXR to CT chest from 6 to 3.6 days[37]. However, as with the Scottish service evaluation, there was extensive pathway redesign within the 'after' pathway, which the authors recognized as a significant limitation in attributing improvements to AI. Storey and colleagues also used a weak reference standard, namely lung cancer suspected on CT chest, rather than the robust reference standard of confirmed lung cancer diagnosis in the current study.

Although some of these studies show more favorable results for the classifiers shown in Table 3, they may not be directly comparable as the CXRs were acquired in a variety of settings, retrospectively selected and in some, enriched to include specific findings[34]. In the present study, cases were unselected and, in the context of primary care, requested a CXR, and were much greater in number. In addition, we were able to show the number of downstream lung cancer diagnoses for each classification.

The LungIMPACT trial has shown that immediate AI prioritization had no impact on important clinical outcomes for patients with suspected lung cancer in the English NHS and that this tool, as assessed, does not currently require potentially time-consuming and costly efforts to include in clinical services. This is likely to apply to the use of any CXR algorithm in this way, because the vast majority of cancers were detected where AI was abnormal and prioritization was therefore tested in a large number of cases. Even modest differences in accuracy are unlikely to change the primary outcome. Furthermore, this finding may extend to other settings where there is a relatively rapid review of images through the usual clinical pathway. Although the study applies directly to the English NHS, some findings may apply to other

healthcare systems with established pathways. Because AI prioritization, although associated with a shortened time to CXR report, did not influence more important clinical outcomes, services would need to implement additional pathway changes to potentially replicate our previous findings. A recommendation could be that any AI-flagged abnormality prompts an immediate human review and, if confirmed, the implementation of a downstream bundle of investigations. The large-scale analysis of secondary outcomes shows the considerable potential burden of FP results, using radiologist and radiographer review as the gold standard, while also showing the considerable number of lung cancers diagnosed in this medium-to-high-risk group. This supports ongoing efforts to better manage these patients.

## Online content

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

Nick Woznitza [1,2], Lesley Smith[3], Janette Rawlinson [4], Iain Au-Yong [5], Bindu George [5], Madava G. Djearaman [6], Arjun Nair[1], Richard W. Lee [7,8,9], Neal Navani [2,10], Siyabonga Ndwandwe[11], Caroline S. Clarke [11], Andrew Creeden[12], Josh Newsome[12], Indrajeet Das[12], Sylvia Abaokporo [5], Richard Tucker[5], James Hathorn[13] & David R. Baldwin [14,15]✉

[1]University College London Hospitals NHS Foundation Trust, London, UK. [2]Lungs for Living Research Centre, University College London, London, UK. [3]Leeds Cancer Research UK Clinical Trials Unit, University of Leeds, Leeds, UK. [4]Tipton, UK. [5]Department of Radiology, Nottingham University Hospitals, Nottingham, UK. [6]Radiology Department, University Hospitals Birmingham NHS Foundation Trust, Birmingham, UK. [7]The Early Diagnosis and Detection Centre, The Royal Marsden NHS Foundation Trust, London, UK. [8]Institute of Cancer Research, London, UK. [9]National Heart and Lung Institute, Imperial College London, London, UK. [10]Department of Thoracic Medicine, University College London Hospital, London, UK. [11]Research Department of Primary Care and Population Health, University College London, London, UK. [12]Radiology Department, University Hospitals of Leicester NHS Trust, Leicester, UK. [13]Radiology Department, East Suffolk and North East Essex NHS Foundation Trust, Colchester, UK. [14]Respiratory Medicine, Nottingham University Hospitals NHS Trust, Nottingham, UK. [15]University of Nottingham, Nottingham, UK. ✉e-mail: david.baldwin12@nhs.net

## Methods

### Study design

A prospective, multicenter, randomized controlled trial was conducted between July 2023 and December 2024 across five NHS Trusts in England. Participants aged 18 years or older, attending for a primary care-requested CXR, were block-randomized by day and site to either immediate AI prioritization of reporting or no AI prioritization. Anteroposterior or posteroanterior views were analyzed and lateral (not routinely performed in the UK) views were excluded.

Pre-allocation to the intervention or routine care was performed using random sampling via a 1:1 randomization method for Monday-to-Friday routine imaging whole-day sessions. Days were randomly allocated to minimize the potential impact of different levels of staffing and capacity. All imaging (CXR or CT) was performed as part of routine care, with the final decision made by the clinicians.

Extended Data Fig. 2 illustrates the pathway and randomization. A single CXR AI (qXR, Qure.ai Technologies) was used in the study. The AI was applied to both arms at the time of image acquisition, so that the reporter had access to the AI-marked images at the time of reporting in both arms of the study.

Extended Data Fig. 2 shows that radiographers could, at their discretion, flag abnormalities that may require further action, which may result in a CT scan, sometimes performed on the same day, which is a preferred option in the NOLCP. Radiographer flags could also be applied to any other (noncancer) findings where they consider that action is potentially needed. The usual clinical pathway was followed where any action was considered to be necessary or optimal. Where there was no flag, the CXR was later reported by a radiologist or reporting radiographer according to local usual clinical practice.

### AI algorithm

qXR (v 4.0, Qure.ai Technologies, India) is a class IIb CE-certified deep learning algorithm already in routine clinical use in some NHS Hospitals. It functions as a clinical decision support tool to assist healthcare professionals in identifying abnormalities (29 classes/types) on CXRs. Images were pseudonymized, transferred to the cloud-based qXR for analysis and then returned to the local Picture Archiving and Communication System with image mark-ups and abnormality classification. On intervention days, an active notification was sent to the worklist for any 'qXR-suspected-abnormal' cases, allowing prioritization for immediate reporting by radiology. The software algorithm did not undergo updates or changes for the duration of the study to ensure consistency.

Training on the use of the AI was provided to all reporters during the implementation of the AI at each hospital and before study commencement. Training covered intended use as a clinical decision support tool, the different AI findings, radiology worklist prioritization and the user interface/AI results.

### Sample size

Based on data from previous work[13], the median time to lung cancer diagnosis was 63 days in the standard reporting group and using a conservative reduction of 10 days, we calculated that 265 cases per group would be needed to detect a difference with 95% power. The expected prevalence of lung cancer in this cohort (primary care referrals for CXR) is 0.6%[13] and thus 100,000 CXRs were judged sufficient (600 cancer cases). For time to CT, the NHS Diagnostic Imaging Dataset records a median of 15 days between request and CT, but data are not segregated between urgent scans and those that are routine or for follow up[20]. The mean and standard deviation of the time between abnormal CXR and CT chest are unknown. Cohen's $d$ (effect size) was used to estimate sample size with power 0.1, 0.3 and 0.5 (considered small, moderate and large effect sizes)[38]. Assuming approximately equal distribution of the number of scans taken, 4,000 CT scans were found to provide adequate power to detect even the smallest effect size (Cohen's $d = 0.1$). A clinically meaningful difference in time from abnormal CXR to CT chest of three days was defined, using the maximum time recommended in the NOLCP.

### Data

Data were collected from existing routine clinical data sources using electronic methods and transferred to a single database. Data verification was performed by manually checking the dates of CT and lung cancer diagnosis by the investigators without knowledge of the study arm. Primary and secondary outcomes with invalid dates that generated negative time intervals were excluded during the data cleaning process.

### Safety and discordance

All CXRs (in both arms) were evaluated for discordance between AI findings and the radiology report. This was done for each of the 29 AI class findings that the AI detected on the CXR, grouped into eight categories based on radiological–clinical relevance and are reported in the results with a ninth category of 'other'. CXRs with a positive AI class finding not taken up for further testing (radiology report normal for that finding) were reviewed. All AI findings that were either potentially serious, such as pneumothorax, or potentially indicative of cancer, such as nodules and hilar enlargement, were also reviewed. CXRs where the only disagreement between AI positive and report negative were unlikely to be suspicious for cancer were capped for pragmatic reasons; cardiomegaly (1,000), blunted costophrenic angle (1,000), pleural effusion (20%) and fibrosis (20%). All reviews were undertaken by local experienced thoracic radiologists or reporting radiographers with the facility for a further opinion.

The primary outcomes were as follows: (1) the difference in time (in days from CXR acquisition) to the diagnosis of patients with lung cancer who had CXRs with AI support at the time of CXR acquisition and prioritization for immediate review and those that had no immediate read but the AI read was available at the time of reporting and (2) the difference in time (in days from CXR acquisition) to CT for patients who had CXRs with AI support at the time of CXR acquisition and prioritization for immediate review and those that had no immediate read but the AI read was available at the time of reporting.

The secondary outcomes were as follows: (1) days to urgent lung cancer referral as defined by the time between CXR acquisition and lung cancer referral (2WW); (2) days to treatment start for patients with lung cancer as defined by the time between CXR acquisition and cancer treatment start date; (3) agreement between qXR and human readers for present/absent findings on CXRs referred from primary care, by each classification; (4) number of urgent lung cancer (2WW) referrals; (5) incidence of lung cancer; (6) stage of lung cancer at diagnosis and (7) algorithm sensitivity, specificity and agreement between report and AI (Kappa).

### Data handling of multiple CXRs per patients

Some patients had multiple CXRs over the study period. For the main analysis of the primary outcomes, we calculated time to CT and time to cancer diagnosis from the first CXR (to avoid duplication and overcounting outcomes). Sensitivity analysis (described below) was also conducted to assess the robustness of this assumption.

### Statistical analysis

Statistical analysis was conducted by an independent statistician (LS) without input from the study investigators. Statistical analyses were performed using Stata/MP 19.5 (StataCorp).

Baseline characteristics of the study population were tabulated using summary statistics overall and by intervention group (immediate AI prioritization or no AI prioritization; Table 1). No statistical testing was carried out on this data.

Graphical representations of the time to lung cancer diagnosis and time to CT scan between the two groups were generated using

histograms and boxplots. As the time distributions for both these outcomes exhibited a right-skewed distribution, a log transformation was used for formal statistical testing. A *t* test on the log transformation of these data was used to test the null hypothesis that mean values on the log-transformed scale are equal. This is equivalent to the null hypothesis that the geometric means on the untransformed scale are equal. The primary outcome, time to CT scan, was further examined by restricting the sample to (1) CT scans within 14 days of CXR or (2) CXRs that were flagged for CT referral only. This was because, in many cases, CTs were performed for other reasons much later, independently of the CXR request, and were unrelated to the hypothesis being tested. On manual review of data, when the time to CT exceeded 14 days, the CT was never triggered by the index CXR randomized for the trial. In addition, some CXRs were coded by radiology to indicate that a CT had been recommended.

Other time-related endpoints (time to urgent referral and time to treatment starting) were also assessed using a *t* test on log-transformed time duration data. The number of urgent referrals, incidence of lung cancer and stage of lung cancer at diagnosis were compared between the two groups using chi-squared tests. The secondary endpoint of agreement between the reporter and the AI was summarized as overall agreement, measured using the kappa statistic.

Subgroup analyses were performed by site and year quarter for both primary endpoints. Sensitivity analyses addressing the issue of multiple CXRs per patient were conducted by identifying the closest CXR before the CT scan as the index CXR or by excluding patients with multiple CXRs.

Further exploratory analysis (not prespecified) of the combined arms was conducted for those diagnosed with cancer, examining only whether both the radiologist and AI CXR reports were normal or abnormal. CXR reports were flagged as normal if all features were recorded as absent; if any feature was recorded as present, the CXR report was classified as abnormal. Comparisons of time to CT scan and time to cancer diagnosis between radiologist and AI CXR reports as normal or abnormal were made using linear regression, with the outcome variables log-transformed as described above.

### Ethical and safety considerations

Favorable ethical approval was obtained from the East of England−Cambridge East Research Ethics Committee (23/EE/0014, 21 February 2023). The study was undertaken with strict adherence to recommended CONSORT guidelines and Good Clinical Practice[39]. The data was held securely, and information governance rules were rigorously followed by all persons involved in the management of the trial protocol or data at the site level, as well as by the investigators.

The study did not directly recruit patients; rather, it evaluated health service delivery. Individual patients were not randomized; rather, their CXRs were. Consent was not obtained from patients, but in each department, clear messaging indicated that AI was being used as part of a research study, and details on how to opt out of the study were provided, as agreed by the Ethics Committee. No patient-identifiable data were available outside the direct clinical care team and data were pseudonymized before analysis.

The health economic analysis will be reported separately. The study protocol is available in the Supplementary Information.

### Reporting summary

Further information on research design is available in the Nature Portfolio Reporting Summary linked to this article.

### Data availability

All data for the study is stored by the study sponsor. Access to fully anonymised data can be requested from the sponsor at: Research & Innovation, Nottingham University Hospitals NHS Trust, Queen's Medical Centre, B Floor, Medical School, Derby Road, Nottingham NG7 2UH, UK. Email: nuhnt.researchsponsor@nhs.net. Data would normally be available within two months of the request.

### Code availability

The code for the AI algorithm is not available, as this, like many others in this field, is a commercial product. The code for the statistical analysis is available from L.S. and has been uploaded to GitHub; see https://github.com/LFairleySmith/LungIMPACT.

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

### Acknowledgements

This work was supported by the Small Business Research Initiative (SBRI) Healthcare (grant SBRIC01P3039 to N.W. and D.R.B.). The funder undertook regular reviews of project progress and funding allocation. We are indebted to the Qure.ai Technologies team for their work on the implementation of the AI and for ensuring all interactions with the NHS ran smoothly and that data were collected in full compliance with data protection requirements. We thank the NHS information technology, research and innovation teams for their help in navigating this relatively new area of research. Qure.ai Technologies had no influence on the study design or analysis. The latter was conducted by a statistician independently of the trial investigators. Qure.ai Technologies assisted in the deployment of qXR, working with the NHS to integrate with local IT systems, ensure pseudonymization and the correct prioritization method, which could be switched on or off according to the randomization.

### Author contributions

N.W. and D.R.B. conceived the study design and wrote the study protocol, obtained ethics approval, maintained oversight of the study and drafted the first version of the manuscript. L.S. designed the statistical analysis plan and conducted all statistical analyses. S.N. and C.S.C. assisted with data cleaning and took part in analytical discussions. J.R. contributed/coordinated PPI work for this study, including the patient panel and part of the study team, including governance. A.N., R.W.L. and N.N. contributed to the study protocol and provided oversight of the study. I.A.Y., B.G., M.G.D., I.D. and J.H. contributed to the study protocol and coordinated data collection at sites. A.C., J.N., S.A. and R.T. performed data collection. N.W., D.R.B., L.S., S.N., C.S.C., A.N., R.W.L. and N.N. edited the final version of the manuscript. I.A.Y., B.G., M.G.D., I.D., J.H., A.C., J.N., S.A. and R.T. reviewed and approved the final version of the manuscript.

### Competing interests

D.R.B. declares honoraria for speaking and education from AstraZeneca and Boehringer Ingelheim, unrelated to the current study. He has received research grants from the National Institute of Health Research, Cancer Research UK, Horizon Europe, Ruth Strauss Foundation, SBRI, Innovate UK, Yorkshire Cancer Research, the Royal Castle Foundation, and UK Research and Innovation. N.W. declares consultancy fees from InHealth, Apollo Radiology (UK) and SMR Health & Tech unrelated to the current study and a travel grant from Qure.ai Technologies to attend the European Congress of Radiology 2023. R.W.L. is funded by the Royal Marsden National Institute for Health and Care Research (NIHR) Biomedical Research Centre and The Royal Marsden Cancer Charity; his institution receives compensation for time spent in a secondment role with NHS England for the Lung Health Check Programme and as National Specialty Lead for the National Institute for Health and Care Research; he has received

research funding from Cancer Research UK, Innovate UK (cofunded by GE Healthcare, Roche Diagnostics, Optellum, Elliptica and RNA Guardian), SBRI (including as a co-applicant with Qure.ai), RM Partners Cancer Alliance and NIHR (including co-applicant on grants with Optellum); he has received honoraria, speaker and advisory fees, and/or hospitality and travel expenses from Cancer Research UK, Roche Diagnostics, Johnson & Johnson, Guardant Health, AstraZeneca and King Faisal Specialist Hospital and Research Centre, Saudi Arabia; and he also undertakes private medical practice. These activities are unrelated to the present study, aside from the SBRI funding received in collaboration with Qure.ai. N.N. is supported by a Medical Research Council Clinical Academic Research Partnership (MR/T02481X/1). N.N. has also received grant funding from the National Institute of Health Research, Cancer Research UK, Horizon Europe, Ruth Strauss Foundation, and the Engineering and Physical Sciences Research Council. This work was partly undertaken at University College London Hospitals NHS Foundation Trust/University College London, which received a proportion of funding from the UK Department of Health's NIHR Biomedical Research Centre's funding scheme. N.N. reports honoraria for nonpromotional educational talks, conference attendance or advisory board participation from Amgen, AstraZeneca, AXANA, BeiGene, Boehringer Ingelheim, Bristol Myers Squibb, EQRx, Fujifilm, Guardant Health, Intuitive, Janssen, Eli Lilly and Company, Merck Sharp & Dohme, Olympus Corporation, Roche and Sanofi. The remaining authors declare no competing interests.

## Additional information

**Extended data** is available for this paper at https://doi.org/10.1038/s41591-026-04253-5.

**Correspondence and requests for materials** should be addressed to David R. Baldwin.

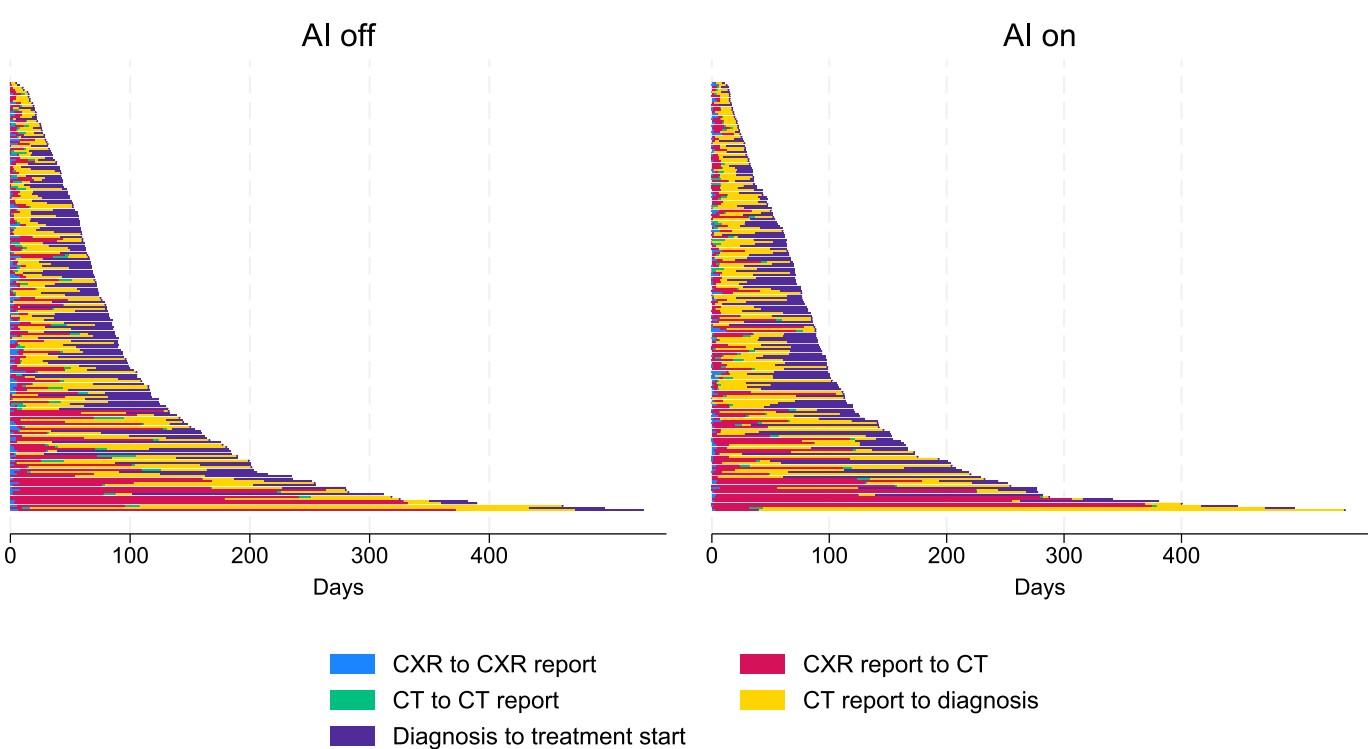

**Extended Data Fig. 1 | Duration on pathway by patient.** This data visualization is limited to those diagnosed with cancer and valid dates for all events presented in the pathway, n = 382, 192 with AI prioritization and 190 with no AI prioritization.

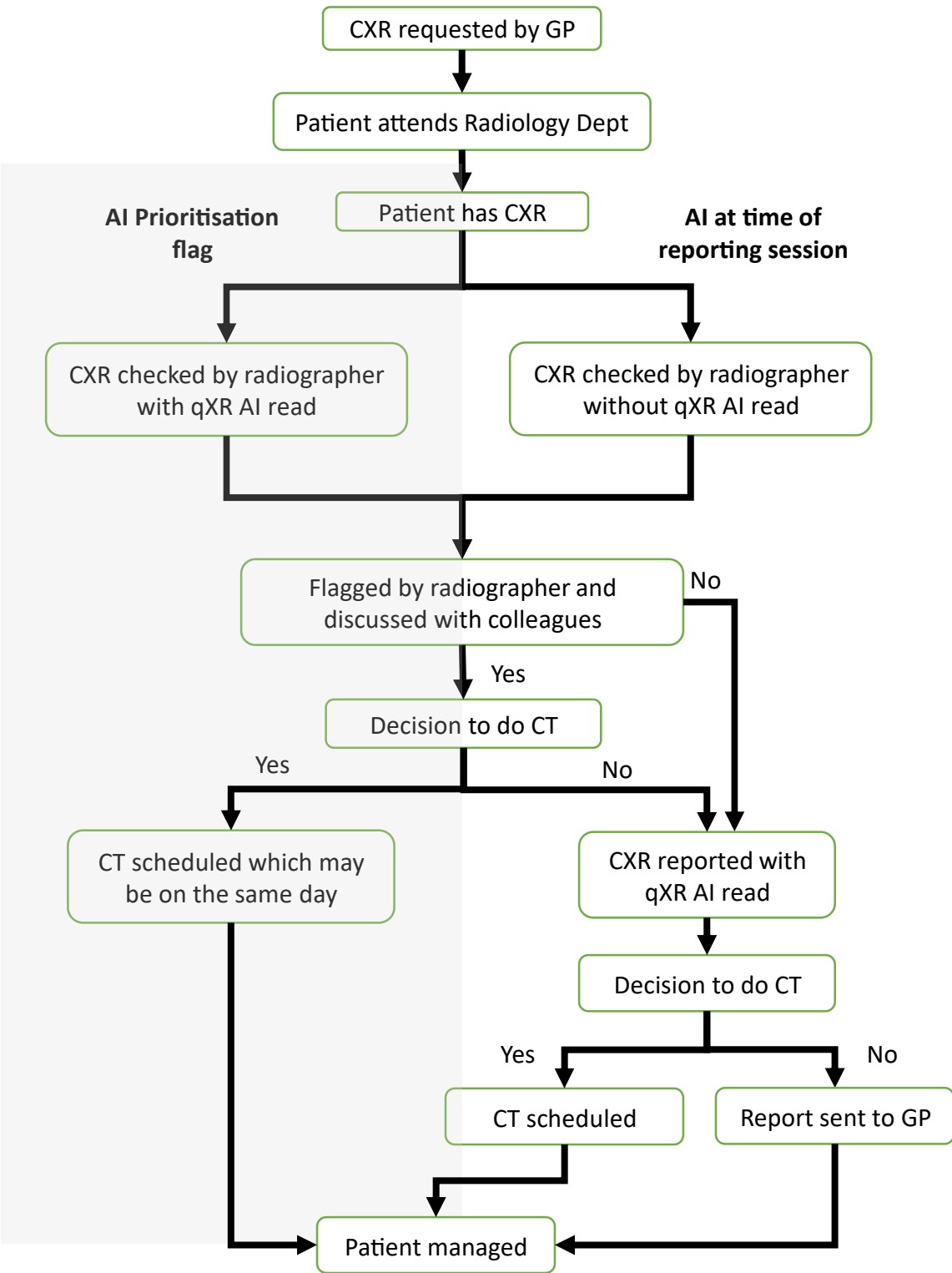

**Extended Data Fig. 2 | Pathway and randomization.** The radiographer may, at their discretion, flag abnormalities that may require further action and this may result in a CT scan being done, sometimes on the same day, which is a preferred option in the NOLCP. The radiographer flag can also happen for any other (noncancer) findings where they consider that action is potentially needed. The usual clinical pathway was followed, where action was confirmed to be necessary or optimal. Where there was no flag, the CXR was later reported by a radiologist or reporting radiographer according to local usual clinical practice.

**Extended Data Table 1 | Primary outcomes stratified by site, ratio of geometric means comparing AI prioritization groups**

| Outcome | Site | N | Ratio of geometric means | 95% CI | P |
|---|---|---|---|---|---|
| **Time to CT scan** | UCLH | 1532 | 1.06 | (0.85, 1.32) | 0.59 |
| | UHL | 7785 | 0.96 | (0.91, 1.02) | 0.16 |
| | ESNEFT | 377 | 0.84 | (0.63, 1.12) | 0.22 |
| | UHB | 1024 | 1.03 | (0.87, 1.22) | 0.72 |
| | NUH | 2629 | 0.97 | (0.86, 1.09) | 0.61 |
| **Time to lung cancer diagnosis** | UCLH | 37 | 1.04 | (0.57, 1.89) | 0.90 |
| | UHL | 323 | 1.08 | (0.87, 1.35) | 0.48 |
| | ESNEFT | 61 | 0.59 | (0.35, 1.00) | 0.05 |
| | UHB | 40 | 0.94 | (0.54, 1.64) | 0.82 |
| | NUH | 97 | 0.98 | (0.67, 1.44) | 0.93 |

Statistical analyses were performed using a two-sided t-test on log-transformed outcomes and presented as the ratio of geometric means and 95% confidence intervals.

**Extended Data Table 2 | Primary outcomes stratified by quarter, ratio of geometric means comparing AI prioritization groups**

| Outcome | Quarter | N | Ratio of geometric means | 95%CI | P |
|---|---|---|---|---|---|
| **Time to CT scan** | Q3 2023 | 270 | 1.06 | (0.62, 1.79) | 0.84 |
| | Q4 2023 | 2214 | 0.89 | (0.79, 1.01) | 0.08 |
| | Q1 2024 | 2657 | 0.95 | (0.85, 1.06) | 0.36 |
| | Q2 2024 | 3631 | 0.91 | (0.85, 1.00) | 0.05 |
| | Q3 2024 | 2535 | 1.14 | (1.02, 1.28) | 0.03 |
| | Q4 2024 | 2040 | 0.98 | (0.87, 1.11) | 0.75 |
| **Time to lung cancer diagnosis** | Q3 2023 | 12 | 1.49 | (0.27, 8.23) | 0.61 |
| | Q4 2023 | 79 | 0.77 | (0.46, 1.28) | 0.31 |
| | Q1 2024 | 124 | 0.94 | (0.66, 1.34) | 0.74 |
| | Q2 2024 | 154 | 1.02 | (0.74, 1.42) | 0.89 |
| | Q3 2024 | 106 | 1.30 | (0.91, 1.86) | 0.15 |
| | Q4 2024 | 83 | 0.71 | (0.50, 1.01) | 0.05 |

Statistical analyses were performed using a two-sided t-test on log-transformed outcomes and presented as the ratio of geometric means and 95% confidence intervals.

**Extended Data Table 3 | Secondary outcomes by AI prioritization**

| | AI prioritisation day | | |
| --- | --- | --- | --- |
| | **No** | **Yes** | |
| | **44,112 (50.7%)*** | **42,833 (49.3%)*** | |
| | **N (%)** | **N (%)** | **P-value** |
| 2WW referral | | | |
| No | 42,828 (97.1%) | 41,618 (97.2%) | 0.51 |
| Yes | 1284 (2.9%) | 1,215 (28%) | |
| Lung cancer diagnosis | | | |
| No | 43,822 (99.3%) | 42,564 (99.4%) | 0.62 |
| Yes | 289 (0.7%) | 269 (0.6%) | |
| *Cancer stage*** | | | |
| *1A* | *11* | *13* | *NA* |
| *1A1* | *5* | *2* | |
| *1A2* | *27* | *11* | |
| *1A3* | *17* | *10* | |
| *1B* | *10* | *16* | |
| *2A* | *3* | *3* | |
| *2B* | *12* | *19* | |
| *3A* | *23* | *26* | |
| *3B* | *29* | *18* | |
| *3C* | *9* | *12* | |
| *4A* | *63* | *68* | |
| *4B* | *38* | *40* | |
| *4C* | *23* | *9* | |
| *Missing* | *19* | *22* | |
| Cancer stage** | | | |
| Stage 1 | 70 (25.9%) | 52 (21.0%) | 0.34 |
| Stage 2 | 15 (5.6%) | 22 (8.9%) | |
| Stage 3 | 61 (22.6%) | 56 (22.7%) | |
| Stage 4 | 124 (45.9%) | 117 (47.4%) | |

*Only incudes first CXR for patients with multiple CXRs in study period

**percentage out of those with cancer diagnosis and stage recorded

Statistical analyses were performed using two-sided chi-squared tests.

**Extended Data Table 4 | Primary outcomes stratified by sex, median and interquartile range and ratio of geometric means comparing AI prioritization groups**

| | AI Prioritisation | | | | | | | | |
|---|---|---|---|---|---|---|---|---|---|
| | No | | | Yes | | | | | |
| | N | Median | IQR | N | Median | IQR | Ratio of GM | 95% CI | P |
| **Time to CT scan (days)** | | | | | | | | | |
| Males | 3358 | 49 | (17, 132) | 3341 | 50 | (17, 139) | 1.01 | (0.94, 1.08) | 0.86 |
| Females | 3315 | 57 | (21, 153) | 3333 | 55 | (18, 153) | 0.94 | (0.88, 1.01) | 0.10 |
| **Time to lung cancer diagnosis (days)** | | | | | | | | | |
| Males | 150 | 49 | (25, 115) | 148 | 41 | (22.5, 83) | 0.88 | (0.70, 1.11) | 0.29 |
| Females | 139 | 44 | (24, 97) | 121 | 50 | (27, 99) | 1.12 | (0.88, 1.41) | 0.36 |

Ratio GM = Ratio of Geometric Means. Statistical analyses were performed using a two-sided t-test on log-transformed outcomes and presented as the ratio of geometric means and 95% confidence intervals.

**Extended Data Table 5 | Agreement between radiologist reports and Qure AI report by abnormality**

| Abnormality | Radiologist report | Qure AI | | Kappa |
| --- | --- | --- | --- | --- |
| | | Absent | Present | |
| Nodule | Absent | 85,014 | 6,845 | 0.15 |
| | Present | 663 | 804 | |
| Cavity | Absent | 92,953 | 259 | 0.22 |
| | Present | 68 | 46 | |
| Mediastinal widening | Absent | 91,723 | 709 | 0.05 |
| | Present | 849 | 45 | |
| Hilar Enlargement | Absent | 89,450 | 3065 | 0.11 |
| | Present | 555 | 256 | |
| Blunted CP/ Pleural effusion | Absent | 86,521 | 3877 | 0.47 |
| | Present | 694 | 2234 | |
| Opacity | Absent | 63,852 | 12,762 | 0.47 |
| | Present | 4590 | 12,122 | |
| Cardiomegaly | Absent | 85,558 | 2429 | 0.48 |
| | Present | 2709 | 2630 | |
| Pneumothorax | Absent | 92,840 | 440 | 0.09 |
| | Present | 22 | 24 | |
| Others | Absent | 86,494 | 2439 | 0.19 |
| | Present | 3531 | 862 | |

Kappa statistic to report levels of agreement between radiologist report and Qure AI report

# Reporting Summary

## Statistics

For all statistical analyses, confirm that the following items are present in the figure legend, table legend, main text, or Methods section.

| n/a | Confirmed | |
|---|---|---|
| ☐ | ☒ | The exact sample size (*n*) for each experimental group/condition, given as a discrete number and unit of measurement |
| ☐ | ☒ | A statement on whether measurements were taken from distinct samples or whether the same sample was measured repeatedly |
| ☐ | ☒ | The statistical test(s) used AND whether they are one- or two-sided *Only common tests should be described solely by name; describe more complex techniques in the Methods section.* |
| ☐ | ☒ | A description of all covariates tested |
| ☐ | ☒ | A description of any assumptions or corrections, such as tests of normality and adjustment for multiple comparisons |
| ☐ | ☒ | A full description of the statistical parameters including central tendency (e.g. means) or other basic estimates (e.g. regression coefficient) AND variation (e.g. standard deviation) or associated estimates of uncertainty (e.g. confidence intervals) |
| ☐ | ☒ | For null hypothesis testing, the test statistic (e.g. *F*, *t*, *r*) with confidence intervals, effect sizes, degrees of freedom and *P* value noted *Give P values as exact values whenever suitable.* |
| ☒ | ☐ | For Bayesian analysis, information on the choice of priors and Markov chain Monte Carlo settings |
| ☒ | ☐ | For hierarchical and complex designs, identification of the appropriate level for tests and full reporting of outcomes |
| ☐ | ☒ | Estimates of effect sizes (e.g. Cohen's *d*, Pearson's *r*), indicating how they were calculated |

*Our web collection on statistics for biologists contains articles on many of the points above.*

## Software and code

Policy information about availability of computer code

| Data collection | qXR (v 4.0, Qure.ai Technologies Limited, UK), is a class IIb CE certified deep learning algorithm and was used as the CXR AI intervention qXR is a proprietary, MDR CE-class IIb commercial product by Qure.ai. Source code and model weights are not open-source and are not publicly downloadable. Data was collected electronically and manually verified for CT data and lung cancer data. |
|---|---|
| Data analysis | All data were analysed with Stata v19.5 (StataCorp, College Station, TX, USA), according to the pre-specified analysis plan. Code has been deposited on GitHib |

For manuscripts utilizing custom algorithms or software that are central to the research but not yet described in published literature, software must be made available to editors and reviewers. We strongly encourage code deposition in a community repository (e.g. GitHub). See the Nature Portfolio guidelines for submitting code & software for further information.

## Data

Policy information about availability of data

All manuscripts must include a data availability statement. This statement should provide the following information, where applicable:
- Accession codes, unique identifiers, or web links for publicly available datasets
- A description of any restrictions on data availability
- For clinical datasets or third party data, please ensure that the statement adheres to our policy

Data were collected and pseudonymised for analysis. The non-consenting nature of the study means that access to the data held by the sponsor requires an additional ethical approval. Access to deannonymised data is only available to the clinical teams at each partcipating site.

## Research involving human participants, their data, or biological material

Policy information about studies with human participants or human data. See also policy information about sex, gender (identity/presentation), and sexual orientation and race, ethnicity and racism.

| | |
|---|---|
| Reporting on sex and gender | Sex data only collected; Female 24,709 (53.7%) in intervention and 25,352 (53.6%) in control. A subgroup analysis was performed and is included in a supplementary upload |
| Reporting on race, ethnicity, or other socially relevant groupings | No data were collected for race, ethnicity or social grouping. |
| Population characteristics | Mean age 59.1 (SD 17.4), same for intervention and control; 53.6% female, same for intervention and control |
| Recruitment | Participants aged 18 years or over, who were attending for a primary care requested CXR, were block randomised by day and site, into immediate AI prioritisation of reporting versus no AI prioritisation. Pre-allocation to intervention or routine care was via a random sampling on a 1:1 randomisation method for Monday to Friday routine imaging whole day sessions. |
| Ethics oversight | East of England Cambridge East REC; 23/EE/0014 21st February 2023 |

Note that full information on the approval of the study protocol must also be provided in the manuscript.

# Field-specific reporting

Please select the one below that is the best fit for your research. If you are not sure, read the appropriate sections before making your selection.

☒ Life sciences  ☐ Behavioural & social sciences  ☐ Ecological, evolutionary & environmental sciences

For a reference copy of the document with all sections, see nature.com/documents/nr-reporting-summary-flat.pdf

# Life sciences study design

All studies must disclose on these points even when the disclosure is negative.

| | |
|---|---|
| Sample size | Using data from previous work,(20) the median time to lung cancer diagnosis was 63 days in the standard reporting group and using a conservative reduction of 10 days, we calculated that 265 cases per group would be needed to detect a difference with 95% power. The expected prevalence of lung cancer in this cohort (primary care referrals for CXR) is 0.6%(20) and thus 100,000 CXRs were judged  sufficient (600 cancer cases).  For time to CT, the NHS Diagnostic Imaging Dataset records a median 15 days between request and CT, but data are not segregated between urgent scans and those that are routine or for follow up(21). The mean and standard deviation of time between abnormal CXR and CT chest are not known.  Cohen's d (effect size) was used to estimate sample size with power 0.1, 0.3 and 0.5 (considered small, moderate and large effect size)(22). Assuming approximately equal distribution of the number of scans taken, 4,000 CT scans were found to provide adequate power to detect even the smallest effect size (Cohen's d = 0.1). A clinically meaningful difference in time from abnormal CXR to CT chest as 3 days was defined, using the maximum time recommended in the NOLCP. |
| Data exclusions | 4,405 CXRs excluded; data compliance issue/technical failure n=1,382, weekend CXR n=3,023 |
| Replication | A major strength of the trial is the randomised controlled design without the need to take individual consent from participants. This meant that there was minimal influence on the normal clinical pathway and more equitable inclusion of groups underrepresented in research. Information in the radiology departments made it clear that there was an option to opt out of the study but there was only one. In addition, the trial was multicentre and included different sizes, geographies and catchment demographics of NHS hospitals. |
| Randomization | Participants aged 18 years or over, who were attending for a primary care requested CXR, were block randomised by day and site, into immediate AI prioritisation of reporting versus no AI prioritisation. Pre-allocation to intervention or routine care was via a random sampling on a 1:1 randomisation method for Monday to Friday routine imaging whole day sessions. |
| Blinding | Blinding was not possible, the intervention was radiology worklist prioritisation. However radiology clinicians were blinded to the randomsiation schedule/sequence. |

# Reporting for specific materials, systems and methods

We require information from authors about some types of materials, experimental systems and methods used in many studies. Here, indicate whether each material, system or method listed is relevant to your study. If you are not sure if a list item applies to your research, read the appropriate section before selecting a response.

## Materials & experimental systems

| n/a | Involved in the study |
|-----|----------------------|
| ☒ | ☐ Antibodies |
| ☒ | ☐ Eukaryotic cell lines |
| ☒ | ☐ Palaeontology and archaeology |
| ☒ | ☐ Animals and other organisms |
| ☐ | ☒ Clinical data |
| ☒ | ☐ Dual use research of concern |
| ☒ | ☐ Plants |

## Methods

| n/a | Involved in the study |
|-----|----------------------|
| ☒ | ☐ ChIP-seq |
| ☒ | ☐ Flow cytometry |
| ☒ | ☐ MRI-based neuroimaging |

## Clinical data

Policy information about clinical studies

All manuscripts should comply with the ICMJE guidelines for publication of clinical research and a completed CONSORT checklist must be included with all submissions.

| | |
|---|---|
| Clinical trial registration | Trial registration ISRCTN 78987039 7th March 2023 |
| Study protocol | Available at trial registration https://www.isrctn.com/ISRCTN78987039 |
| Data collection | Between 17th July 2023 and 31st December 2024, 97,731 CXRs were performed across five diverse NHS Trusts, with a geographical spread and a mix of general/tertiary and high/low volume centres. Patients were followed up until 5th June 2025. The mean age of the study population was 59 years and 46% were male. University Hospitals of Leicester (UHL) performed 54% of the study CXRs, Nottingham University Hospitals (NUH) 15%, East Sussex and North Essex Foundation Trust (ESNEFT) 11%, University College London Hospital (UCLH) 11% and University Hospitals of Birmingham (UHB) 9% (Table S1) |
| Outcomes | The primary outcomes were:<br>1. The difference in time (in days from CXR request) to the diagnosis of lung cancer for patients who have CXRs with AI support at the time of CXR acquisition and prioritisation for immediate review and those that have no immediate read but the AI read is available at the time of reporting.<br>2. The difference in time (in days from CXR request) to CT for patients who have CXRs with AI support at the time of CXR acquisition and prioritisation for immediate review and those that have no immediate read but the AI read is available at the time of reporting.<br>The secondary outcomes were:<br>1. Days to urgent lung cancer referral as defined by time between CXR acquisition and lung cancer referral (2WW)<br>2. Days to treatment start for lung cancer patients<br>3. Agreement between qXR and human readers for present/absent findings on CXRs referred from primary care, by each classification<br>4. Number of urgent lung cancer (2WW) referrals<br>5. Incidence of lung cancer<br>6. Stage of lung cancer at diagnosis<br>7. Algorithm sensitivity, specificity and agreement between report and AI (Kappa) |

## Plants

| | |
|---|---|
| Seed stocks | *Report on the source of all seed stocks or other plant material used. If applicable, state the seed stock centre and catalogue number. If plant specimens were collected from the field, describe the collection location, date and sampling procedures.* |
| Novel plant genotypes | *Describe the methods by which all novel plant genotypes were produced. This includes those generated by transgenic approaches, gene editing, chemical/radiation-based mutagenesis and hybridization. For transgenic lines, describe the transformation method, the number of independent lines analyzed and the generation upon which experiments were performed. For gene-edited lines, describe the editor used, the endogenous sequence targeted for editing, the targeting guide RNA sequence (if applicable) and how the editor was applied.* |
| Authentication | *Describe any authentication procedures for each seed stock used or novel genotype generated. Describe any experiments used to assess the effect of a mutation and, where applicable, how potential secondary effects (e.g. second site T-DNA insertions, mosiacism, off-target gene editing) were examined.* |

