## [Peer Review File · Nature Medicine]

AI-based chest X-ray prioritization in the lung cancer diagnostic pathway: The LungIMPACT randomized controlled trial

Corresponding Author: Professor David Baldwin

Version 0:

Reviewer comments:

Reviewer #2

(Remarks to the Author)

Major / General Comments

The study considers prioritization of CT following a chest X-ray that is read by a radiologist and an AI system. The goal of the study is to quantify whether such prioritization leads to shorter times to CT acquisition and lung cancer diagnosis. It is unclear why and how the use of AI is relevant as this study seems more of an analysis of the prioritization strategy rather than the utility of an AI tool. In other words, it is unclear if the AI system should be expected to change management in any way (perhaps the study design could be better explained). Moreover, the results show that such “prioritization strategy” leads to no significant difference in the primary outcomes. The motivation for the study may be to achieve the NOLCP mandate of progressing suspicious CXR to CT within 72 hours or to validate the reduction in time to lung cancer diagnosis reported previously (reference 13). Clearly this is not achieved, and the authors conclude that the complexity of the workflow is not justified. This is true regarding how AI was used in this study, but is not sufficient to rule out its use as a diagnostic or predictive tool when implemented in a way that can actually affect the clinical workflow. Given the study design, the results are not entirely surprising and do not provide a strong argument for how they will impact clinical care.

Minor / Technical Comments

- The performance / utility of the model used is not provided. Why would the use of this AI help or harm the study?
- The study design is not clearly laid out at the beginning. Explicitly highlight that this is not a test of AI vs no AI, but of AI prioritization vs AI-without-prioritization.
- The primary analyses rely on t-tests of log-transformed time intervals, equivalent to testing equality of geometric means. Why is the difference in geometric means the right statistic to consider?
- The discussion focuses on the lack of benefit but less on possible harms (for instance, increase reading workload?).
- The primary end points seem to be defined in days from CXR request, and the time to CT is greater is observed to be greater than time to diagnosis – this seems counter-intuitive but is not discussed.

Reviewer #3

(Remarks to the Author)

The LungIMPACT study is a large, multicenter, block-randomized controlled trial evaluating the impact of immediate AI prioritization of chest X-rays (CXRs) on the lung cancer diagnostic pathway. The statistical analysis plan (SAP) and manuscript are well-aligned, and the study employs appropriate methods for its design and data characteristics. The manuscript reports both primary and secondary endpoints as outlined in the SAP. These include time to CT scan and time to lung cancer diagnosis, as well as agreement metrics between AI and human readers. Appropriate Statistical Methods were used for data analysis: Use of log-transformed t-tests for skewed time-to-event data is methodologically sound. Chi-squared tests are suitable for categorical comparisons, and Kappa statistics are appropriate for agreement analyses. Subgroup and sensitivity analyses are well-conceived and executed.

However, several statistical limitations merit attention.

1. The discordance between AI and radiologist interpretations yielded statistically and clinically meaningful insights. These findings are underemphasized and could be more prominently featured in the abstract and discussion.
2. While pragmatic, day-level randomization may introduce confounding (e.g., weekday effects, staffing variability). The manuscript should provide a more thorough justification for this choice and discuss its implications for internal validity.
3. Although a separate analysis is planned, even a brief summary of expected cost implications would strengthen the manuscript, especially given the resource investment required for AI implementation.

Reviewer #4

(Remarks to the Author)

This large, multi-centre randomized controlled trial evaluated whether immediate AI-based prioritisation of chest X-rays requested from primary care could speed up the lung cancer diagnostic pathway compared with usual reporting. The results are straightforward: the AI tool may have some diagnostic value, but prioritising images based on AI did not improve the lung-cancer pathway.

Even though this is a large, carefully run study, there are several issues that limit the strength and broader relevance of the findings:

- 1) The manuscript does not dig into the reasons AI prioritisation failed. Was it CT scanner capacity? heavy workload in the NHS? workflow bottlenecks elsewhere? Without this analysis, the message feels incomplete
- 2) Because the AI markings were shown to readers in both arms, the study is essentially comparing “AI with a prioritisation alert” vs “AI without the alert.” This weakens the intervention and makes it much harder to detect a meaningful effect
- 3) The UK uses chest X-ray heavily as a first-line test for respiratory symptoms, and the workflow (radiographer flags, CT scheduling delays, referral patterns) is very specific to the NHS. Many health systems would not see the same issues or timelines. This limits the international relevance.
- 4) It tells us many AI flags are false positives, and some cancers appear in AI-positive but radiologist-normal cases. But the study doesn't use these findings to propose how clinicians should actually act on AI results.

Reviewer #5

(Remarks to the Author)

This manuscript presents results from the LungIMPACT randomized controlled trial evaluating whether AI-based CXR prioritization (qXR) accelerates time to chest CT or time to lung cancer diagnosis among patients referred from primary care. The study is large, multicentre, and prospectively randomized by day. The authors report no difference in either primary outcome.

The study addresses an important evidence gap. However, several key issues related to workflow implementation, analytic decisions, and reader blinding require clarification before the findings can be fully interpreted. In its current form, the manuscript does not demonstrate that the intervention—AI-driven prioritization—was implemented in a way that could plausibly influence the outcomes being measured. These issues substantially affect interpretability and generalizability.

Major Comments

1. The trial workflow appears insufficient to allow AI prioritization to influence outcomes

The manuscript does not specify what proportion of AI-flagged CXRs were reviewed before the patient left the department, a prerequisite for facilitating same-day CT or rapid CT booking according to the National Optimal Lung Cancer Pathway (NOLCP). If radiographer or radiologist reports were routinely issued within 24 hours, as in the authors' previous single center study (Woznitza et al., Thorax 2023), AI prioritization may not meaningfully accelerate downstream care.

The median time to CT was 53 days in both arms, far exceeding the NOLCP ≤ 72 -hour benchmark. This strongly suggests that (a) patients commonly left before any expedited action could occur, or (b) CT capacity constraints dominated the pathway. Under such conditions, AI prioritization is not capable of influencing the primary outcomes, regardless of AI algorithm performance.

Requests for clarification:

- What proportion of AI-flagged CXRs were reviewed before the patient left the imaging department?
- What were the radiology reporting turnaround times across all sites?
- How frequently was same-day or ≤ 72 -hour CT performed for AI-flagged cases?
- Were capacity constraints (particularly CT availability) documented?

Without this operational detail, the negative primary results may reflect structural pathway limitations rather than lack of AI utility.

2. Justification for restricting analyses to CTs performed within 14 days is unclear

The rationale for restricting some analyses to CTs within 14 days of CXR is not clearly explained. Given that the median time to CT was 53 days, this subset represents a small, highly selected group and may not reflect routine practice. It is unclear whether this restriction was preregistered, and how it affects interpretation. Restricting analysis to early CTs

risks introducing selection bias and reduces alignment with the clinical question (whether AI prioritization accelerates the real-world lung cancer pathway).

3. Lack of radiologist blinding to AI outputs prevents evaluation of added value

AI outputs were available to readers in both arms, which makes it impossible to determine:

- what AI would have identified independently,
- what radiologists would have identified without AI assistance, and
- whether “missed” findings were due to AI, or human readers.

This severely limits interpretation of discordance analyses, miss rates, and any conclusions about safety or incremental diagnostic benefit.

The manuscript should clearly acknowledge this limitation and clarify how discordance classification was adjudicated in a non-blinded setting.

Version 1:

Reviewer comments:

Reviewer #2

(Remarks to the Author)

Reviewer #3

(Remarks to the Author)

Many results are referenced in another work due to word count limitations.

Reviewer #4

(Remarks to the Author)

The major concern of this study is a mismatch between the research question and the trial design. Although the authors aim to evaluate whether AI-driven prioritisation can accelerate the lung cancer diagnostic pathway, AI results were available in both study arms. Only the prioritisation “switch” differed. As a consequence, the intervention has a minimal and indirect effect on clinical workflow, making a null result highly predictable.

Moreover, AI-positive but radiologist-negative cases were systematically reviewed and sometimes led to further diagnostic actions. This means that AI influenced clinical decision-making in both groups, reducing the contrast between intervention and control and limiting the ability of the study to detect any real effect.

The primary outcomes were also diluted by including a large proportion of CT scans that were not triggered by the index CXR. Given that the median time to CT was over 50 days, the study likely captures noise from downstream clinical decisions unrelated to the prioritisation mechanism.

Overall, the trial evaluates the operational value of an AI “priority flag” rather than the clinical impact of AI-assisted interpretation. Because system bottlenecks (e.g., CT capacity and pathway organisation) remain unchanged, prioritisation alone is unlikely to influence time to CT or diagnosis. Therefore, while the negative findings are valid, they mainly reflect limitations in study design and health-system constraints rather than the ineffectiveness of AI.

Reviewer #5

(Remarks to the Author)

Thank you for providing additional information on the study.

1. The median time from CXR to reporting was 34.1 hours in the AI group versus 47 hours in the no AI group. With an average 13 hours difference, AI was set up to fail and the primary objective of this study cannot be properly evaluated. If the study design was similar to the pilot study, AI would at least have a fair chance. The appropriate study design should have been having the patient with an abnormal CXR flagged by AI wait until a radiologist had reviewed the CXR before he/she left the radiology department. If the radiologist felt a chest CT was indicated, one could be done on the same day or an appointment given to return on another day for the CT. According to Supplemental Table S8, there were only 672 patients suspected to have lung cancer who were referred for a CT. During the 18 months study, this amounted to an average of 37 CTs a month distributed in 5 sites. It seems highly doable to have reserved spots for a CT on the same day for patients with abnormal CXR flagged by AI that the radiologist concurred one should be done.

2. In page 9, the authors responded that 172 CT scans were completed on the same day as the CXR: 88 in the AI prioritization arm and 84 in the no AI arm. How did it come about in the no AI prioritization arm, some patients had their CXR read before they left? This would make it even more difficult to evaluate the role of AI prioritization.

3. In the consort diagram, it would be good to indicate the number of CXRs flagged by AI or no AI flags and the number of

lung cancers in each group.

4. Training of the radiologists to familiarize them with the AI and an agreed reporting protocol should be in place prior to the study. This would address lack of trust from false positives findings such as cavities, hilar enlargement, cardiomegaly or even pneumothorax. A common reporting format would also improve the acceptance of the CXR reports by the primary care providers or referring physicians to take action on the next step such as requesting a CT scan for suspected lung cancer to speed up the lung cancer care pathway.

5. Since the AI report was available to the radiologist, one can only compare the accuracy of AI versus radiologist + AI. If detection of CXRs with findings suspicious of lung cancer were the goal, it would be more informative to compare the sensitivity and specificity of AI versus radiologist + AI. The detailed breakdown of the abnormalities in Supplemental Table S7 and the analyses in Supplemental Table S11 would not be necessary or belong to another paper as suggested by the authors.

Reviewer	Comment	Response	Page
Editor			
E1	A more detailed discussion of the differences between the two arms	This has been done and figure 2 (previously figure 1 before methods moved down as per request E25 below) referenced again. The title of the paper has also been changed to help with this.	Title, method and discussion
E2	potential explanations for the findings	We have expanded on this	discussion
E3	a clear acknowledgment of the study's limitations	We have made this section clearer	Discussion
E4	any post hoc analysis is clearly signposted	This is now, hopefully, clear	Discussion
E5 Abstract	should be no more than 200 words and for trials should adhere to the CONSORT framework.	Now 200 words and conforming to CONSORT content	Page 4
E6 Abstract	primary outcome was or was not met	Clearly stated	Page 4
E7 abstract	provide the effect size and relevant uncertainty estimate	This is now provided	Page 4
E8 abstract	conclusion of the study must focus only on the primary outcome and safety/tolerability	Now includes only the joint primary outcomes	Page 4
E9 abstract	You must report either all or none of the secondary outcomes, given the space limitation, I would suggest removing all mention of secondary outcomes from the abstract	Insufficient space for this so all removed as suggested. However, we hope this has not detracted from the useful observations from the secondary outcomes	Page 4
E10 abstract	Please include the trial registration number at the end of the Abstract	Included	Page 4
E11	Introduction should be written for a broad, non-specialist medical reader and provide sufficient context for the work.	We feel that the current background section complies with this but if there are specific areas that are not clear we would be happy to update. We have clarified how our previous RCT relates to the current study.	Page 5
E12	Please provide details of your cohort in the first paragraph of the Results. This	This is described	Page 6

	includes number of individuals screened for enrolment, as well as exact dates of first and last patient enrolment		
E13	The CONSORT patient disposition diagram and the baseline characteristics must be included as main non-text items (for which there is a strict limit of 6 tables and/or figures)	Not applicable, already met	N/A
E14	The Results should be structured as followed:  - Patient disposition - Primary outcome(s) - Secondary outcomes - Safety - Exploratory outcomes - Sensitivity analyses - Post-hoc analyses 	Patient disposition is not relevant to this study We have clearly divided the results according to the sections stipulated – mostly this was already compliant.	Pages 6-9
E15	Please remove any subheadings from the Discussion	Removed	Pages 10-14
E16	Please ensure all results are presented in the Results section, no new data should be introduced in the Discussion	No new data are in the discussion	Page 6-9
E17	You must include explicit paragraphs of study limitations in the Discussion	These are included	Pages 10-14
E18	The overarching conclusion of the study must be based only on the primary outcome and safety data	This now only includes the primary outcomes and safety (discordance) data	Page 14
E19	The Methods should include a full description of the inclusion and exclusion criteria, as well as study procedures and statistical analyses (including a power calculation)	Already included	Pages 24-29

E20	Please upload the protocol and SAP with the revision materials, so that reviewers and editors have access to them	Uploaded	N/A
E21	You must ensure that contributions from all individuals in the author list are available in the Author Contributions statement	We are compliant with this	Page 15
E22	Please move all funding sources to the Acknowledgements, including a statement on the role of the funder	Complete	Page 15
E23	Please ensure that all potential competing interests are detailed for all authors. For any authors with no competing interests, this must also be stated	Already included	Page 15
E24	Please see our guidelines for the Data and Code Availability Statements. "Available on request" is not acceptable, you must provide details of any restrictions to data and code availability. https://www.nature.com/nature-portfolio/editorial-policies/reporting-standards#availability-of-data	Provided	Page 29
E25	The article file must only contain these items in this order:  • Title • Author List and affiliations • Abstract • Introduction • Results (with Subheadings) • Discussion • Acknowledgements • Author Contributions • Competing Interests Statement 	Provided in order	

	 • References (for main text only) • Figure legends (for main text only) • Tables (note: tables should be pasted into Word files as editable tables, not as images) • Methods • Data Availability Statement • Code Availability Statement • Methods-only References 		
Reviewer 2	Thank you for your detailed review		
R2.1	It is unclear why and how the use of AI is relevant as this study seems more of an analysis of the prioritization strategy rather than the utility of an AI tool.	The primary outcome is only concerned with the prioritisation function of AI. The secondary outcomes were developed to quantify aspects of performance of the tool. This has been made clear and is explicit on Figure 2 (previously figure 1)	Page 24-25
R2.2	it is unclear if the AI system should be expected to change management in any way (perhaps the study design could be better explained)	We have attempted to clarify this further but feel that it is clear, especially if one looks at figure 2 (previously figure 1). The primary outcomes were impact of AI prioritisation on two key measures of speed in the lung cancer pathway. As stated on page 12, this would require individual consent, which would then massively disrupt the normal workflow. The secondary outcomes were more concerned with aspects of the diagnostic performance of the algorithm. Our study has removed the influence of this aspect, which may vary between products, on the primary outcome.	Page 12, fig 2 and methods
R2.3	This is true regarding how AI was used in this study, but is not sufficient to rule out its use as a diagnostic or predictive tool when	We have not suggested that the use of AI is ruled out. We have shown that AI prioritisation, in this context, does not impact on important clinical	Page 12,

	implemented in a way that can actually affect the clinical workflow. Given the study design, the results are not entirely surprising and do not provide a strong argument for how they will impact clinical care.	outcomes in lung cancer and have suggested this applies to other similar areas. Although this study presents the results of a large randomised trial that show there is no effect of prioritisation, this could not be predicted as our previous study had shown such a large impact of radiographer triage on time to lung cancer diagnosis. We are only able to discuss the merits of including AI as a tool to improve accuracy and our discordance analysis quantifies not only how much these tools have to be used with a great deal of human interpretation, but also the frequency of lung cancer diagnosis according to human and AI result. The study also shows that an expensive and potentially disruptive element of AI products is not required in this context. These points have already been covered in the discussion.	
R2.4	The performance / utility of the model used is not provided. Why would the use of this AI help or harm the study?	This study has quantified the level of discordance with this tool, which will likely apply to others. This does not argue against their use but does show what can be expected and the extent of human interpretation that will be needed to use them effectively. A further study will look in detail at the downstream outcomes and how this relates to human and AI interpretation.	Pages 12-14
R2.5	The study design is not clearly laid out at the beginning. Explicitly highlight that this is not a test of AI vs no AI, but of AI prioritization vs AI-without-prioritization.	This has been clarified somewhat but we refer the reviewer to figure 2 (previously figure 1 as methods are now after the references), which we created during the initial ethics application in order to clarify exactly the point the reviewer is making.	Page 25
R2.6	The primary analyses rely on t-tests of log-transformed time intervals, equivalent to	See Review 3 (statistician) comments: Appropriate Statistical Methods were used for data analysis:	

	testing equality of geometric means. Why is the difference in geometric means the right statistic to consider?	Use of log-transformed t-tests for skewed time-to-event data is methodologically sound. Chi-squared tests are suitable for categorical comparisons, and Kappa statistics are appropriate for agreement analyses. Subgroup and sensitivity analyses are well-conceived and executed	
R2.7	The discussion focuses on the lack of benefit but less on possible harms (for instance, increase reading workload?)	We have added more on the potential for workload disruption, which was already covered in the discussion on page 10. Specifically, we have added to a sentence in the discussion noting that this aspect will depend on the way the user uses the AI and knowledge of the strengths and weaknesses. This is apparent in, for example use of CAD for pulmonary nodules, where radiologists learn where the CAD is reliable and where it is not. Further work (as stated) is required to go into the detail of the relatively few cases where AI findings were dismissed by the reporter and cancer was diagnosed downstream. This was not part of this study, as we would have been unable to estimate a sample size.	Page 10 para 3 Page 11 Para 2
R2.8	The primary end points seem to be defined in days from CXR request, and the time to CT is greater is observed to be greater than time to diagnosis – this seems counter-intuitive but is not discussed.	We have added to the paragraph in the discussion that covered this and addressed directly this point to avoid confusion. The primary analysis was for any CT downstream of the CXR acquisition . We found this was often unrelated to the primary episode that generated the CXR. Thus, we also analysed the data according to whether the CT had been triggered by the CXR and also restricted the data to CT within 14 days (to pick up like related CTs, e.g. normal CXR to CT pathways). All analyses showed	Page 13 Para 1

		no difference, so we are very confident of this result. We hope the text explains this better.	
Reviewer 3	Thank you for your detailed review		
R3.1	The discordance between AI and radiologist interpretations yielded statistically and clinically meaningful insights. These findings are underemphasized and could be more prominently featured in the abstract and discussion.	See Editorial comment E9 that suggests removal of all secondary outcomes from the abstract We would like to include these but hope the reader will be able to appreciate them in the main paper. With editorial permission for extra words in the abstract we could include them.	Page 4
R3.2	While pragmatic, day-level randomization may introduce confounding (e.g., weekday effects, staffing variability). The manuscript should provide a more thorough justification for this choice and discuss its implications for internal validity.	The randomisation did not use specific days – these were randomly distributed without preference for any particular day in each arm. This should minimise any impact of day level variability on the primary outcomes. We have made this clear in the methods section.	Pages 24-29
R3.3	Although a separate analysis is planned, even a brief summary of expected cost implications would strengthen the manuscript, especially given the resource investment required for AI implementation.	We agree that this would add to the manuscript, but we have a detail analysis that is to be submitted separately. We have added a very brief pointer to the excess cost and its avoidability on page 21	Page13 Para 4
Reviewer 4	Thank you for the detailed review		
R4.1	The manuscript does not dig into the reasons AI prioritisation failed. Was it CT scanner capacity? heavy workload in the NHS? workflow bottlenecks elsewhere? Without this analysis, the message feels incomplete	Thank you for this observation. We note that the time from CXR acquisition to report in the 5 NHS Trusts participating was short, in both arms. We now also present in table 2, the times from CXR acquisition to report. The reason for prioritisation failing was that the median time was 34 hours in the	Page 22 Page 11 para 1

		prioritisation group and 47 in the non prioritisation. This meant that the AI did not trigger the same immediate action that we saw in our previous study. We have now included a section in the discussion developing this point	
R4.2	Because the AI markings were shown to readers in both arms, the study is essentially comparing “AI with a prioritisation alert” vs “AI without the alert.” This weakens the intervention and makes it much harder to detect a meaningful effect	See also R2.5 and R5.12 Although we agree that very accurate AI could mean better triage on the pathway, we do not consider this to be the main way in which AI would impact the primary outcome of time to CT and thus does not weaken the intervention. The reason for the joint primary outcomes was precisely for this point. We have added some text to expand this point	Page 25 Page 11
R4.3	The UK uses chest X-ray heavily as a first-line test for respiratory symptoms, and the workflow (radiographer flags, CT scheduling delays, referral patterns) is very specific to the NHS. Many health systems would not see the same issues or timelines. This limits the international relevance.	We agree that this is an NHS study but we also think that this applies to any service where there is an established workflow template (rather than one that is totally unregulated). The study has shown that AI prioritisation alone has no impact and so clinical teams need to ensure that the pathway includes modifications not related to the AI but that may be based around the use. An example would be mandating immediate human review and an downstream bundle of tests based on that. This would then approximate what was done in our former study. We have developed this in the discussion.	Page 14
R4.4	It tells us many AI flags are false positives, and some cancers appear in AI-positive but radiologist-normal cases. But the study	We agree but this requires a detailed review of all findings (including non-lung cancer) this is the subject of a further research project that may be highly informative	Page 14 para 2

	doesn't use these findings to propose how clinicians should actually act on AI results.		
Reviewer 5	Thank you for your detailed review.		
R5.1	The manuscript does not specify what proportion of AI-flagged CXRs were reviewed before the patient left the department, a prerequisite for facilitating same-day CT or rapid CT booking according to the National Optimal Lung Cancer Pathway (NOLCP).	This has now been included in the results. There were 88 in the prioritisation arm and 84 in the non prioritisation arm	Page 9
R5.2	If radiographer or radiologist reports were routinely issued within 24 hours, as in the authors' previous single center study (Woznitza et al., Thorax 2023), AI prioritization may not meaningfully accelerate downstream care.	The report issue may not be the same as action on the finding – please see response to point 4.1 and additional text on page 11.	Page 22 Page 11 para 1
R5.3	The median time to CT was 53 days in both arms, far exceeding the NOLCP ≤72-hour benchmark.	We have clarified the explanation for this and the need for further sensitivity analysis by CTs with 14 days and CXR to CT code. The primary analysis was for any CT downstream of the CXR acquisition . We found this was often unrelated to the primary episode that generated the CXR. Thus, we also analysed the data according to whether the CT had been triggered by the CXR and also restricted the data to CT within 14 days (to pick up like related CTs, e.g. normal CXR to CT pathways). The fact that all show no difference provides us with confidence in the conclusions.	Page 13
R5.4	What proportion of AI-flagged CXRs were reviewed before the patient left the imaging department?	Please see response above	

R5.5	What were the radiology reporting turnaround times across all sites?	This is now included in table 2, and does show significantly shorter times for AI prioritisation (p<0.0001)	Page 12
R5.6	How frequently was same-day or ≤72-hour CT performed for AI-flagged cases?	This has been added. There were 477 CT scans within the maximum time of 72 hours from CXR as set out in the NOLCP, with 241 in the prioritisation on and 236 in the prioritisation off arms	Page 9
R5.7	Were capacity constraints (particularly CT availability) documented?	See also R4.1 We did not document this but have added some text in the discussion on page 22	Page 11 Para 1
R5.8	Justification for restricting analyses to CTs performed within 14 days is unclear It is unclear whether this restriction was preregistered, and how it affects interpretation. Restricting analysis to early CTs risks introducing selection bias and reduces alignment with the clinical question (whether AI prioritization accelerates the real-world lung cancer pathway).	Prior to the study we did not know the extent of downstream CT usage – as you can see this was very considerable. By direct review of the CT reports (a lot of work) we found that no CTs done after 14 days were directly triggered by the CXR. We therefore felt a sensitivity analysis was necessary by both CXR to CT code and CT within 14 days. The fact that all are negative strengthens the overall conclusion that prioritisation does not accelerate the pathway and additional changes need to be mandated (already recommended in NOLCP) to have any impact.	Pages 11-12 para 1
R5.9	Lack of radiologist blinding to AI outputs prevents evaluation of added value	We were not testing the added value of AI in this study but the huge number of discordance reviews merits and further analysis which is intended over the next year.	Page 13 para 2
R5.10	AI outputs were available to readers in both arms, which makes it impossible to determine: •what AI would have identified independently, • what radiologists would have identified	See also R2.3 comment We agree that we are not able to identify these and we have added text in the discussion to acknowledge this. However, this was not the intention of the study and would have required a different method. The discordance review was	Page 13, para 2 Page 11 para 2

	without AI assistance, and  • whether “missed” findings were due to AI, or human readers. 	simply to examine where reporter and AI differed and to quantify the extent of this. This means that the discordance was with the benefit of the AI mark up. The purpose of the review was to confirm the discordance, which in the vast majority was confirmed but in a small number, as stated, there were cases where the reporter had either missed a finding or dismissed one that was clinically significant. This is not presented in this paper because it requires and detail case review at each of the participating sites and is one of the planned follow-on studies	
R5.11	This severely limits interpretation of discordance analyses, miss rates, and any conclusions about safety or incremental diagnostic benefit.	See also E3 and E17 comments. This is a limitation of the study, but we do have the opportunity to provide more information about this through careful analysis of the downstream outcomes of the 28K discordance reviews	Page 13
R5.12	The manuscript should clearly acknowledge this limitation and clarify how discordance classification was adjudicated in a non-blinded setting.	This has been included	Page 13

Reviewer	Comment	Response
Editor	All comments	Dealt with in last revision including formatting and word counts
E1	If the research findings apply to only one sex or gender, that must be indicated in the title and/or abstract.	Not applicable
E2a	For studies involving vertebrates animal and cell lines- The Reporting Summary should include whether sex was were considered in the study design.	Not applicable
E2b	For studies involving human research participants- The Reporting Summary should include whether sex and/or gender was considered in the study design and whether sex and/or gender of participants was determined based on self-report or assigned (and methodology used).	Sex data was collected from existing electronic health records, no additional data were collected for the study
E3	Data should be reported disaggregated for sex and gender where this information has been collected and consent has been obtained for reporting and sharing individual-level data; disaggregated numbers for individual experiments must be provided in the source data as appropriate whereas overall numbers may be provided in the Nature Portfolio Reporting Summary.	Sex data was collected from existing electronic health records, no additional data were collected for the study
Reviewer 4		

		Many thanks for your detailed re-review and comments
4.1	The major concern of this study is a mismatch between the research question and the trial design. Although the authors aim to evaluate whether AI-driven prioritisation can accelerate the lung cancer diagnostic pathway, AI results were available in both study arms. Only the prioritisation “switch” differed. As a consequence, the intervention has a minimal and indirect effect on clinical workflow, making a null result highly predictable.	Thank you for your comment, which emphasises the primary outcome of the trial – we were investigating if AI triage (not the availability of AI) would impact on the time to CT or lung cancer diagnosis. However, for reasons previously stated and included in the last revision of the manuscript, we do not agree that the results was highly predictable. This is easy to say in retrospect with the results of a large RCT to hand. Indeed, our previous study showed a huge effect of radiographer triage given the same constraints in NHS services. Thus, we needed to know if the triage element (prioritisation) of AI would have a similar effect or not. If it had, it would be a strong argument for implementation of AI and if not, it would argue against the use of AI prioritisation, which is both expensive and potentially disruptive without benefit. We have added a sentence on page 11, para 1

4.2	Moreover, AI-positive but radiologist-negative cases were systematically reviewed and sometimes led to further diagnostic actions. This means that AI influenced clinical decision-making in both groups, reducing the contrast between intervention and control and limiting the ability of the study to detect any real effect.	To clarify this point, the primary endpoints were not influenced by the discordance reviews. Time to lung cancer diagnosis and time to CT were recorded as they happened but the discordance reviews were later. However, we agree that AI is likely to have influence decision making in both groups. This would be important so that we can measure the effect of prioritisation rather than having the AI or not. The latter is more influenced by the accuracy of the AI, which will likely change with newer versions and between products
4.3	The primary outcomes were also diluted by including a large proportion of CT scans that were not triggered by the index CXR. Given that the median time to CT was over 50 days, the study likely captures noise from downstream clinical decisions unrelated to the prioritisation mechanism.	We were concerned to measure all downstream activity as prior to the study we did not know how much of this would be related to the index CXR. This was the reason for the sensitivity analyses which removed or reduce the noise that the reviewer highlights. Our subgroup analysis found that for the CXRs coded for an urgent chest CT (n=1,000) there was no difference between AI prioritisation and no prioritisation Ratio of Geometric Means 1.02 (0.81, 1.28) p=0.86

4.4	the trial evaluates the operational value of an AI “priority flag” rather than the clinical impact of AI-assisted interpretation.	As you have identified above, you have correctly stated that the primary outcome was the impact of AI triage of CXRs on the time to CT and diagnosis of lung cancer. The clinical impact of AI-assisted interpretation is highly dependent on the accuracy of the AI. This is better evaluated in studies with established ground truth. It was not our intention. If AI prioritisation had made a substantial difference to the pathway times then this would have been a very strong argument for immediate implementation of prioritisation.
4.5	Because system bottlenecks (e.g., CT capacity and pathway organisation) remain unchanged, prioritisation alone is unlikely to influence time to CT or diagnosis. Therefore, while the negative findings are valid, they mainly reflect limitations in study design and health-system constraints rather than the ineffectiveness of AI.	See response to 4.3 Many AI studies also involve pathway changes, which have additional financial, resource and workforce costs often not captured or reported. Indeed, a recent service review from Scotland for another AI product could not distinguish the possible benefits from an AI to those wider pathway changes (navigator, additional reporting capacity). We had already included this within the discussion “Our resource impact analysis of the diagnostic pathway found that AI-assisted clinician review of CXRs incurred additional costs compared with the traditional radiology pathway, based on

		pathway changes as part of NHS Grampian’s service evaluation.” https://shtg.scot/media/2526/2024-07-18-chest-x-ray-assessment-v14.pdf An AI tool cannot be introduced in isolation, which is why determining that the other pathway changes identified as bottlenecks by the reviewer are indeed valid but will limit AI triage within the NHS.
Reviewer 5		
		Many thanks for your detailed re-review and comments
Thank you for providing additional information on the study.		Thank you
5.1a	The median time from CXR to reporting was 34.1 hours in the AI group versus 47 hours in the no AI group. With an average 13 hours difference, AI was set up to fail and the primary objective of this study cannot be properly evaluated. If the study design was similar to the pilot study, AI would at least have a fair chance. The appropriate study design should have been having the patient with an abnormal CXR flagged by AI wait until a radiologist had reviewed the CXR before he/she left the radiology department. If the radiologist felt a chest CT was indicated, one could be	See 4.3, 4.5 above Although we do not agree that AI was set up to fail, this point is important and allows us to clarify here and in a further sentence in the discussion. The primary objective has been achieved in that we conclusively show that AI driven prioritisation has no impact on important clinical outcomes, and does not show the same impact as radiographer-driven triage did in our previous study. The study design suggested would have been a pathway change and that on its own, without AI

	done on the same day or an appointment given to return on another day for the CT	prioritisation may have had the effect of reducing pathway time. It should also be noted that the design includes the option to look at the CXR immediately after acquisition, with or without the prioritisation (figure 2) We have added a sentence in the discussion page 11 para 1 Our results suggest that, as an isolated intervention, AI triage of CXRs referred from primary care is unlikely to confer any patient benefit. Further work, as part of a clustered RCT (AI + pathway optimisation vs standard of care) is required.
5.1b	According to Supplemental Table S8, there were only 672 patients suspected to have lung cancer who were referred for a CT. During the 18 months study, this amounted to an average of 37 CTs a month distributed in 5 sites. It seems highly doable to have reserved spots for a CT on the same day for patients with abnormal CXR flagged by AI that the radiologist concurred one should be done.	See 4.3, 4.5 and 5.1a above S8 reports the outcome of the expert review of the 28,261 CXRs with a discrepancy between AI and report. Table 3 reports the total number of chest CT (13,247) with 2,766 performed within 14 days and 1,000 coded for an urgent chest CT. We have changed the title of table S8 to avoid any confusion. It now reads :

		“Suggested outcome after expert discordance review” We agree that having reserved CT appointments would be an excellent approach and very doable. Indeed, this is recommended in guidelines and in the NOCLP. Our study aim was to determine the benefit of AI prioritisation alone, independent of any further pathway redesign or optimisation. To date, we are the only trial that has reported on an AI alone initiative, a significant strength of the current work.
5.2	In page 9, the authors responded that 172 CT scans were completed on the same day as the CXR: 88 in the AI prioritization arm and 84 in the no AI arm. How did it come about in the no AI prioritization arm, some patients had their CXR read before they left? This would make it even more difficult to evaluate the role of AI prioritization.	The need to review CXRs in both arms was addressed early on in the study design. Please refer to figure 2 where this is illustrated and the text immediately beneath in on page 26 which reads “Figure 2 shows that the radiographer may, at their discretion, flag abnormalities that may require further action and this may result in a CT scan being done, sometimes on the same day as is a preferred option in the NOLCP. The radiographer flag can also happen for any other (non-cancer) findings where they consider that action is potentially needed. The usual clinical pathway was followed where action was confirmed to be necessary or optimal...” Furthermore, in the UK radiographers have a professional

		duty/obligation to review all images for serious findings https://www.hcpc-uk.org/standards/standards-of-proficiency/radiographers/ That said, we know that radiology departments are very busy and the hypothesis was that the prioritisation would prompt action more often compared with standard practice. This was not the case.
5.3	In the consort diagram, it would be good to indicate the number of CXRs flagged by AI or no AI flags and the number of lung cancers in each group.	The number of lung cancers within each arm are already within the CONSORT diagram (Figure 1, final box; AI with prioritisation n=269, AI without prioritisation n=289) We have included the number in each arm where AI detected an abnormality
5.4a	Training of the radiologists to familiarize them with the AI and an agreed reporting protocol should be in place prior to the study. This would address lack of trust from false positives findings such as cavities, hilar enlargement, cardiomegaly or even pneumothorax.	Training for all reporters occurred prior to all AI deployments, apologies for this omission from the manuscript. We have included details within the methods (p26 para 1) Training on the use of the AI was provided to all reporters during the implementation of the AI at each hospital and prior to study commencement. Training covered intended use as a clinical decision support

		tool, the different AI findings, radiology worklist prioritisation and the user interface/AI results. Trustworthiness in/of AI is a highly relevant topic, one that merits further work.
5.4b	A common reporting format would also improve the acceptance of the CXR reports by the primary care providers or referring physicians to take action on the next step such as requesting a CT scan for suspected lung cancer to speed up the lung cancer care pathway.	This is an important point and one that could be adopted as part of a CXR to CT pathway. Structured reporting templates for primary care was outside of the scope of this RCT. Many departments have immediate escalation for CT scanning where a CXR is suspicious for lung cancer. Indeed, in this study 1000 had a CXR coded to generate a CT referral. However, AI prioritisation in this subset did not show any difference.
5.5	Since the AI report was available to the radiologist, one can only compare the accuracy of AI versus radiologist + AI. If detection of CXRs with findings suspicious of lung cancer were the goal, it would be more informative to compare the sensitivity and specificity of AI versus radiologist + AI. The detailed breakdown of the abnormalities in Supplemental Table S7 and the analyses in Supplemental Table S11 would not be necessary or belong to	We agree with the logic but the reality is that AI detects a lot of abnormalities that are easily dismissed by the radiologist. Indeed, as these tools are used more, reporters become familiar with the common false positives. More concerning are cases where an AI finding is dismissed when a true positive. Although this applies to relatively few cases, we plan to study these images in more detail and publish our work on this. This may allow a guide to be developed to mitigate the risk of

	another paper as suggested by the authors.	dismissing valid findings. We already discuss this on page 11 para 3. We have not deleted table S7 or S 11 as we believe tha additional detail will be o of interest. Furthermore the statistical reviewer (Reviewer 3) stated all analyses were appropriate.
--	--	---